

# Thermodynamic limit of the two-spinon form factors for the zero field XXX chain

**Nikolai Kitanine[1,2*] and Giridhar Kulkarni [1]**

**1** Institut de Mathématiques de Bourgogne, UMR-CNRS 5584,
Université de Bourgogne, 21078 Dijon, France
**2** Laboratoire de Physique Théorique et Hautes Energies,
UMR 7589, Sorbonne Universités et CNRS, 75005 Paris, France

⋆ Nikolai.Kitanine@u-bourgogne.fr

## Abstract

In this paper we propose a method based on the algebraic Bethe ansatz leading to explicit results for the form factors of quantum spin chains in the thermodynamic limit. Starting from the determinant representations we retrieve in particular the formula for the two-spinon form factors for the isotropic XXX Heisenberg chain obtained initially in the framework of the $q$-vertex operator approach.

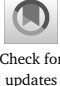

# 1 Introduction

The computation of form factors for integrable quantum field theories [1] and lattice models [2,3] has always been one of the most important and challenging problems of the theory of quantum integrable systems. It gained even more importance since extremely good predictions for neutron scattering experiments were produced by the numerical analysis of the dynamical structure factors based on explicit analytic results for the form factors obtained from the Algebraic Bethe ansatz [4–6] or $q$-vertex operator approach [7,8]. Form factor analysis in the thermodynamic limit leads also to a very powerful method of asymptotic computation of the correlation functions and dynamical structure factors [9–11]. Recently it was also shown that the form factor approach is an excellent tool to compute the correlation function at finite temperature [12].

There are two main methods leading to the explicit evaluation of the form factors for the spin chains: multiple integral representation from the $q$-vertex operator approach obtained by M. Jimbo and T. Miwa [2], and determinant representations [3] obtained using the Algebraic Bethe ansatz [13], determinant formulas for scalar product [14] and the quantum inverse problem for the spin chains [3,15]. In the context of the XXX spin chains, while the first method permits the computation of the form factors for two [16,17] and four-spinon [7,18] excited states for zero external field, the second method has only been applied to the non-zero field case where it lead to the asymptotic analysis of the spin-spin correlation functions.

Quite curiously, the results for the form factors obtained through the $q$-vertex operator approach were never reproduced in the Algebraic Bethe Ansatz (ABA) framework (with only one important exception of the Baxter formula for the spontaneous magnetisation of the XXZ spin chain in the massive regime [19]). While the multiple integral representations for the correlation functions [20,21] were reproduced and generalized to the non-zero magnetic field case [22], the two types of representations for the form factors remained unrelated (apart from the numerical comparison [23]). Typically, the final results for the form factors in the ABA framework [23–25] always contained some Fredholm determinants which could not be expressed as multiple integrals.

The main goal of the present article is to introduce a new approach which allows us to relate the two types of results from these different approaches in the zero external field case. There are several reasons why this relation is extremely important. First of all, the treatment of form factors in the algebraic Bethe ansatz formalism is very similar in different regimes of the XXZ chain including the isotropic limit : XXX chain. In particular, the computations in the massless regime are much more sophisticated in the framework of the $q$-vertex operator approach [26] and based on the results for the elliptic case [27]. Second the Algebraic Bethe ansatz permits to understand the role of the complex roots of the Bethe equations (bound states). Computation of the form factors with bound states for non-zero field were performed in [28] and the final result includes as usual some Fredholm determinants. An approach based on BJSMT fermionic formalism to the bound states was proposed in [29]. We believe that the direct computation from the algebraic Bethe ansatz in the zero field case leads to much more explicit results for the form factors in the thermodynamic limit. And finally we hope that this approach will permit us to go beyond the two and four-spinon cases.

In this paper we compute the matrix elements of local spin operators between the ground state and low-lying excited states (with a small number of spinons which correspond to the number of holes in the Bethe picture). We show that the form factors can be reduced to finite

dimensional determinants (dimension of the remaining matrix being the number of spinons). We illustrate our approach with the simplest case: the two-spinon form factor of the XXX chain. We show in particular that using the Algebraic Bethe ansatz approach, one can obtain the same result as the q-vertex operator technique [16] for these form factors.

It is important to mention that all the results concerning the excited states of the massless spin chains in the Bethe ansatz framework in the zero-field case are based on the assumption that the tails of distribution of Bethe roots do not contribute to the leading order of measurable quantities in the thermodynamic limit. This assumption is already implicitly used in [30, 31] and its particular case, the condensation property of Bethe roots [32, 33] was very recently proved [34]. For the computation of form factors this "bulk assumption" becomes even more important and more difficult to prove. Hence the result for the two-spinon case is extremely important as it permits us to check the validity of the method.

The paper is organised as follows. In the section 2 we recall the basic framework of the Algebraic Bethe ansatz approach [13] and the determinant representations for the norms [35–37], scalar products [14] and form factors [3]. The general framework of our method permitting the computation of the two-spinon form-factor as determinants, as well as its thermodynamic limit, is described in the section 3. Some technical and computational difficulties are addressed in the appendices A and B.

# 2 Form factors from the Algebraic Bethe ansatz

## 2.1 Algebraic Bethe ansatz

We consider here the spin-1/2 isotropic Heisenberg spin chain [38],

$$H = \sum_{m=1}^{M} \left\{ \sigma_m^x \sigma_{m+1}^x + \sigma_m^y \sigma_{m+1}^y + (\sigma_m^z \sigma_{m+1}^z - 1) \right\}, \tag{2.1}$$

with periodic boundary conditions $\sigma_{M+1}^a = \sigma_1^a$. The number of sites $M$ is taken to be even. This operator acts in the *quantum space* $\mathcal{H}_q$ of the model which is, in the case of XXX chain, given by the tensor product of $M$ local quantum spaces $V_m = \mathbb{C}^2$ spaces, $\mathcal{H}_q = V_1 \otimes V_2 \otimes \cdots \otimes V_M$. The eigenstates of this Hamiltonian were first constructed by H. Bethe [39], however here we will follow the algebraic Bethe ansatz approach [13].

We start from the rational solution of the Yang-Baxter equation $R(\lambda)$. It is a $4 \times 4$ matrix acting in a tensor product of two $\mathbb{C}^2$ spaces $V_1 \otimes V_2$

$$R_{12}(\lambda) = \frac{1}{\lambda + i}(\lambda I_{12} + iP_{12}), \tag{2.2}$$

where $I_{12}$ is the identity matrix and $P_{12}$ is the permutation matrix in this tensor product $P_{12}(x \otimes y) = y \otimes x$. We define the monodromy matrix acting in the tensor product of $M + 1$ $\mathbb{C}^2$ spaces $V_0 \otimes \mathcal{H}_q$ (where $V_0$ is called auxiliary space) as an ordered product of $R$ matrices.

$$T_0(\lambda) = R_{0M}\left(\lambda - \frac{i}{2}\right) \ldots R_{01}\left(\lambda - \frac{i}{2}\right) = \begin{pmatrix} A(\lambda) & B(\lambda) \\ C(\lambda) & D(\lambda) \end{pmatrix}_0. \tag{2.3}$$

Here operators $A(\lambda)$, $B(\lambda)$, $C(\lambda)$ and $D(\lambda)$ act in the quantum space $\mathcal{H}_q$ and their commutation relations follow from the Yang-Baxter algebra

$$R_{ab}(\lambda - \mu)T_a(\lambda)T_b(\mu) = T_b(\mu)T_a(\lambda)R_{ab}(\lambda - \mu). \tag{2.4}$$

One immediate corollary of this relation is the fact that the transfer matrix defined as the trace of the monodromy matrix commutes for different values of spectral parameter

$$\mathcal{T}(\lambda) = \text{tr}_0 T(\lambda) = A(\lambda) + D(\lambda), \qquad [\mathcal{T}(\lambda), \mathcal{T}(\mu)] = 0. \tag{2.5}$$

It means in particular that the transfer matrix is a generator of a commuting family of conserved charges as the Hamiltonian can be reconstructed in terms of the logarithmic derivative of the transfer matrix,

$$H = 2i \, \mathcal{T}^{-1}(\lambda) \frac{d}{d\lambda} \mathcal{T}(\lambda) \Big|_{\lambda = \frac{i}{2}}, \qquad [H, \mathcal{T}(\lambda)] = 0. \tag{2.6}$$

In the algebraic Bethe ansatz framework the operators $B(\lambda)$ and $C(\lambda)$ are used as the creation and annihilation operators. Starting from the ferromagnetic state (all the spins up)

$$|0\rangle = \begin{pmatrix} 1 \\ 0 \end{pmatrix} \otimes \cdots \otimes \begin{pmatrix} 1 \\ 0 \end{pmatrix},$$

it was shown in [13] that the states

$$|\Psi(\{\lambda_1, \ldots, \lambda_N\})\rangle = B(\lambda_1) \ldots B(\lambda_N) |0\rangle \tag{2.7}$$

are eigenstates of the transfer matrix (and hence of the Hamiltonian) provided the parameters $\lambda_j$ satisfy the Bethe equations

$$\left( \frac{\lambda_j - \frac{i}{2}}{\lambda_j + \frac{i}{2}} \right)^M \prod_{k=1}^{N} \frac{\lambda_j - \lambda_k + i}{\lambda_j - \lambda_k - i} = -1, \qquad j = 1, \ldots, N. \tag{2.8}$$

We will call vectors $|\Psi(\{\lambda\})\rangle$ on-shell Bethe vectors if (2.8) is satisfied and off-shell Bethe vectors otherwise.

Here we introduce some notations that will be used throughout this paper. For every solution of Bethe equations $\{\lambda_j, j = 1, \ldots, N\}$ we introduce corresponding Baxter polynomial $q(\lambda)$ and exponential counting function $\mathfrak{a}(\lambda)$ as

$$q(\lambda) = \prod_{j=1}^{N} (\lambda - \lambda_j), \qquad \mathfrak{a}(\lambda) = \left( \frac{\lambda - \frac{i}{2}}{\lambda + \frac{i}{2}} \right)^M \frac{q(\lambda + i)}{q(\lambda - i)}. \tag{2.9}$$

Note that in these notations the Bethe equations (2.8) take a simple form $\mathfrak{a}(\lambda_j) + 1 = 0$. The corresponding eigenvalue of the transfer matrix is also expressed in terms of these functions as

$$\mathcal{T}(\mu) |\Psi(\{\lambda\})\rangle = \tau(\mu) |\Psi(\{\lambda\})\rangle, \qquad \tau(\mu) = \left( \mathfrak{a}(\mu) + 1 \right) \frac{q(\mu - i)}{q(\mu)}. \tag{2.10}$$

Corresponding dual vectors for any solution of Bethe equations can be constructed using operators $C(\lambda)$ as

$$\langle \Psi(\{\lambda\})| = \langle 0| C(\lambda_1) \ldots C(\lambda_N), \qquad \langle \Psi(\{\lambda\})| \mathcal{T}(\mu) = \tau(\mu) \langle \Psi(\{\lambda\})|. \tag{2.11}$$

It is important to mention that not all the eigenstates of the XXX chain can be constructed as on-shell Bethe vectors. The particularity of the XXX case is its additional $\mathfrak{su}(2)$ symmetry. It was shown in [31] that the on-shell Bethe states are $\mathfrak{su}(2)$ highest weight vectors

$$S^+ |\Psi(\{\lambda\})\rangle = 0, \qquad S^+ = \sum_{m=1}^{M} \sigma_m^+. \tag{2.12}$$

It means, in particular, that there are no solutions of Bethe equations with $N > \frac{M}{2}$. The Bethe vectors with $N < \frac{M}{2}$ give rise to $M - 2N + 1$ multiplets

$$|\Psi_\ell(\{\lambda\})\rangle = S^{-\ell} |\Psi(\{\lambda\})\rangle, \quad \ell = 0, \ldots, M - 2N. \tag{2.13}$$

All these states are eigenstates of the transfer matrix sharing the same eigenvalue. Note that these states can be obtained as limits of usual off-shell Bethe states using the asymptotic behaviour of the operator $B(\lambda)$

$$\lim_{\lambda \to \infty} \lambda B(\lambda) = iS^-. \tag{2.14}$$

## 2.2 Ground state and excitations

In this subsection we will give a short description of the solutions of Bethe equations corresponding to the ground state and elementary excitations around the ground state. It is well known [30,31] that the ground state $|\Psi_g\rangle$ of the XXX chain is the only Bethe state with $\frac{M}{2}$ real Bethe roots. It means in particular that it is a $\mathfrak{su}(2)$ singlet. In what follows, the Bethe roots corresponding to the ground state will be denoted $\lambda_1, \ldots, \lambda_{\frac{M}{2}}$ with corresponding functions $q_g(\lambda)$, $\mathfrak{a}_g(\lambda)$ and $\tau_g(\lambda)$. The main property of the ground state which distinguishes it from any excited state is the fact that the only real zeroes of the function $\mathfrak{a}_g(\lambda) + 1$ are the Bethe roots (in particular it means that there are no holes).

The Bethe roots are distributed over the real axis in the thermodynamic limit with a density $\rho(\lambda)$

$$\rho(\lambda) = \lim_{M \to \infty} \frac{1}{2\pi i M} \frac{d}{d\lambda} \log\big(\mathfrak{a}_g(\lambda)\big). \tag{2.15}$$

This density can be computed from the Lieb equation

$$\rho(\lambda) + \frac{1}{2\pi i} \int_{-\infty}^{\infty} K(\lambda - \mu)\rho(\mu)d\mu = \frac{1}{2\pi i} t(\lambda - i/2), \tag{2.16}$$

where we have introduced the following notations which will be used throughout the paper.

$$t(\lambda) = \frac{i}{\lambda(\lambda + i)}, \qquad K(\lambda) = \frac{2i}{(\lambda - i)(\lambda + i)} = t(\lambda) + t(-\lambda). \tag{2.17}$$

The solution of this equation has a simple form

$$\rho(\lambda) = \frac{1}{2 \cosh(\pi\lambda)}. \tag{2.18}$$

For any sufficiently regular function $f(\lambda)$ the sums over Bethe roots can be rewritten as integrals with this density in the thermodynamic limit without $1/M$ corrections [32–34]

$$\frac{1}{M} \sum_{j=1}^{\frac{M}{2}} f(\lambda_j) = \int_{-\infty}^{\infty} f(\lambda)\rho(\lambda)d\lambda + o\left(\frac{1}{M}\right). \tag{2.19}$$

The excited states close to the ground states [30, 31, 40] can be constructed introducing an even number of holes and complex roots (in conjugated pairs). The complex roots do not influence energy and momenta of the excited states, however they cannot be neglected at the level of form factors.

For the excited states, the roots will be denoted by $\mu_1, \ldots, \mu_N$ ($N$ can be less than $\frac{M}{2}$ due to the multiplet structure), corresponding Baxter polynomial is $q_e(\lambda)$ and counting function is $\mathfrak{a}_e(\lambda)$. We suppose that there are $N - 2n_c$ real roots and $2n_c$ complex roots (as the complex roots always appear in conjugated pairs). The function $\mathfrak{a}_e(\lambda) + 1$ has real simple zeroes in the positions of the real roots but also $n_h$ additional simple real zeroes corresponding to holes $\mu_{h_1}, \ldots, \mu_{h_{n_h}}$. The number of holes $n_h$ is always even. Determining the position of complex roots is a more complicated question. Following [40,41], we suppose that they form 2-strings, quartets and wide pairs $2n_c = 2n_s + 4n_q + 2n_w$ and their positions satisfy the higher level Bethe equations. This assumption is not crucial for the present paper as we don't treat the complex roots here.

To be more precise, the two-spinon sector treated in this paper consists of two types of states: singlet states with 2 holes and one 2-string, and triplet states with 2 holes and no complex roots [30].

The two-spinon singlet states $|\Psi(\mu_{h_1}, \mu_{h_2} | \mu_c)\rangle$ have $\frac{M}{2} - 2$ real roots, two holes $\mu_{h_1}$ and $\mu_{h_2}$ and two complex roots $\mu_c \pm i(\frac{1}{2} + \delta)$, where position of the string center is fixed $\mu_c = \frac{1}{2}(\mu_{h_1} + \mu_{h_2})$ and string deviation is exponentially small in the thermodynamic limit $\delta = O(M^{-\infty})$.

The two-spinon triplet states $|\Psi_\ell(\mu_{h_1}, \mu_{h_2})\rangle$, $\ell = 0, 1, 2$ do not include any complex roots, there are $\frac{M}{2} - 1$ real roots and 2 holes $\mu_{h_1}$ and $\mu_{h_2}$. Note that the holes also satisfy the equation $1 + \mathfrak{a}(\lambda)$ although they are not counted among these real roots. The total number of real zeroes of the equation $1 + \mathfrak{a}(\lambda)$ is called the occupancy number of the state which is, in this case, $\frac{M}{2} + 1$.

For such excitations parametrised by holes and complex roots, the presence of these parameters add their contribution of order $1/M$ to the density function. For example for the 2-spinon triplet states (which is essential to us in the context of this paper) sums over the roots of $\mathfrak{a}_e(\lambda) + 1$ for any sufficiently regular function $f(\lambda)$ can be written as integrals (condensation property of Bethe roots)

$$\frac{1}{M}\left(\sum_{j=1}^{\frac{M}{2}-1} f(\mu_j) + f(\mu_{h_1}) + f(\mu_{h_2})\right) =$$
$$\int_{-\infty}^{\infty} f(\lambda)\left(\rho(\lambda) + \frac{1}{M}(\rho_h(\lambda - \mu_{h_1}) + \rho_h(\lambda - \mu_{h_2}))\right) d\lambda + o\left(\frac{1}{M}\right), \quad (2.20)$$

where additional density terms satisfy the following integral equation.

$$\rho_h(\lambda) + \frac{1}{2\pi i}\int_{-\infty}^{\infty} K(\lambda - \mu)\rho_h(\mu)d\mu = \frac{1}{2\pi i}K(\lambda). \quad (2.21)$$

This can be generalised to write the density function expansion as the ground state density in the leading order followed by $1/M$ density terms due to the holes and complex roots [40].

The energies and momenta of the excited states do not depend on the complex roots, and can be expressed only in terms of the hole positions, for example the energy and momentum of any two-spinon state (with respect to the ground state) can be written as follows [30],

$$\Delta E \equiv E_e - E_g = \varepsilon(\mu_{h_1}) + \varepsilon(\mu_{h_2}), \qquad \varepsilon(\mu) = \frac{\pi}{2\cosh \pi\mu}, \quad (2.22)$$

$$\Delta P \equiv P_e - P_g = p(\mu_{h_1}) + p(\mu_{h_2}), \qquad p(\mu) = \frac{\pi}{2} - \arctan(\sinh \pi\mu). \quad (2.23)$$

In this paper we will also need the ratios of $q$-polynomials and transfer matrix eigenvalues for the ground and excited states

$$\phi(\lambda) = \frac{q_e(\lambda)}{q_g(\lambda)}, \qquad \chi(\lambda) = \frac{\tau_e(\lambda)}{\tau_g(\lambda)}. \tag{2.24}$$

For the excited state given by two-spinon triplets, the thermodynamic limit of these functions is computed in Appendix B. The thermodynamic limit of the ratio of Baxter polynomials is well defined outside the real axis,

$$\phi(\lambda) = \begin{cases} \frac{1}{2i} \displaystyle\prod_{a=1}^{2} \frac{\Gamma\left(\frac{\lambda - \mu_{h_a}}{2i}\right)}{\Gamma\left(\frac{1}{2} + \frac{\lambda - \mu_{h_a}}{2i}\right)}; & \Im(\lambda) > 0 \\[4mm] -\frac{1}{2i} \displaystyle\prod_{a=1}^{2} \frac{\Gamma\left(-\frac{\lambda - \mu_{h_a}}{2i}\right)}{\Gamma\left(\frac{1}{2} - \frac{\lambda - \mu_{h_a}}{2i}\right)}; & \Im(\lambda) < 0 \end{cases}. \tag{2.25}$$

For the ratio of transfer matrix eigenvalues we get

$$\chi(\lambda) = \prod_{a=1}^{2} \tanh\left(\frac{\pi(\lambda - \mu_{h_a})}{2}\right). \tag{2.26}$$

Let us remark that although we restricted our computations in the appendix B to the two-spinon triplet states, the thermodynamic limit of these quantities $\phi(\lambda)$ and $\chi(\lambda)$ can be easily generalised to other type of excitations.

## 2.3 Scalar products and form factors

The computation of form factors in the framework of the algebraic Bethe ansatz consists of two essential steps: solution of the quantum inverse problem [3] and computation of the scalar products [14].

The main goal of this paper is to compute the following square matrix elements of a local spin operator between the ground state and an excited state

$$|\mathcal{F}_z|^2 = \frac{\langle \Psi_e \, | \, \sigma_m^z \, | \, \Psi_g \rangle \langle \Psi_g \, | \, \sigma_m^z \, | \, \Psi_e \rangle}{\langle \Psi_g \, | \, \Psi_g \rangle \langle \Psi_e \, | \, \Psi_e \rangle}. \tag{2.27}$$

For the XXX chain it is sufficient to consider only $\sigma_m^z$ form factors due to the symmetry of the model.

It was shown in [3] that the local spin operators can be represented in terms of the monodromy matrix elements

$$\sigma_m^z = \mathcal{T}^{m-1}\left(\tfrac{i}{2}\right)\left\{A\left(\tfrac{i}{2}\right) - D\left(\tfrac{i}{2}\right)\right\}\mathcal{T}^{-m}\left(\tfrac{i}{2}\right), \tag{2.28}$$

$$\sigma_m^- = \mathcal{T}^{m-1}\left(\tfrac{i}{2}\right)B\left(\tfrac{i}{2}\right)\mathcal{T}^{-m}\left(\tfrac{i}{2}\right), \tag{2.29}$$

$$\sigma_m^+ = \mathcal{T}^{m-1}\left(\tfrac{i}{2}\right)C\left(\tfrac{i}{2}\right)\mathcal{T}^{-m}\left(\tfrac{i}{2}\right). \tag{2.30}$$

It means that the form factors (2.27) can be reduced to scalar products of on-shell and off-shell Bethe vectors. These scalar products are given by the Slavnov formula [14]. Let $\{\lambda_1, \ldots \lambda_N\}$ be a solution of Bethe equations, $q(\lambda)$ and $\mathfrak{a}(\lambda)$ corresponding functions and $\{\mu_1, \ldots \mu_N\}$ a

generic set of complex parameters. Then the scalar product of the corresponding states can be written as the following determinant

$$\langle\Psi(\{\mu\})|\Psi(\{\lambda\})\rangle = \frac{\prod_{k=1}^{N} q(\mu_k - i)}{\prod_{j>k}(\lambda_j - \lambda_k)(\mu_k - \mu_j)} \det_N \mathcal{M}(\{\lambda\}|\{\mu\}),$$

$$\mathcal{M}_{j,k}(\{\lambda\}|\{\mu\}) = \mathfrak{a}(\mu_k)t(\mu_k - \lambda_j) - t(\lambda_j - \mu_k). \tag{2.31}$$

Whereas the norms of the Bethe states are given by the Gaudin formula [35,36]

$$\langle\Psi(\{\lambda\})|\Psi(\{\lambda\})\rangle = (-1)^N \frac{\prod_{j=1}^{N} q(\lambda_j - i)}{\prod_{j\neq k}(\lambda_j - \lambda_k)} \det \mathcal{N}(\{\lambda\}),$$

$$\mathcal{N}_{j,k}(\{\lambda\}) = \mathfrak{a}'(\lambda_j)\delta_{j,k} - K(\lambda_j - \lambda_k). \tag{2.32}$$

We will now apply these results to the form-factors in the two-spinon sector. One simple remark is in order here. Since the ground state is a singlet, it follows from the relation $\sigma_m^z = [S^+, \sigma_m^-]$ and the highest weight property of the Bethe states (2.12) that the matrix elements of $\sigma_m^z$ between two singlets are zero. As a result, all the non-zero form factors in the two-spinon sector comes from the triplet excitations

$$|\mathcal{F}_z(\mu_{h_1}, \mu_{h_2})|^2 = \frac{\langle\Psi_1(\mu_{h_1}, \mu_{h_2})|\sigma_m^z|\Psi_g\rangle \langle\Psi_g|\sigma_m^z|\Psi_1(\mu_{h_1}, \mu_{h_2})\rangle}{\langle\Psi_g|\Psi_g\rangle \langle\Psi_1(\mu_{h_1}, \mu_{h_2})|\Psi_1(\mu_{h_1}, \mu_{h_2})\rangle}. \tag{2.33}$$

Here we remark that this is true for any number of spinons since with the similar arguments, one can also show that the matrix elements of $\sigma_m^z$ between a singlet and any other multiplet which is not triplet are zero.

To make the computation of form-factors more straightforward, here we use a Foda-Wheeler version of the Slavnov determinant formula for the multiplet Bethe states obtained in [42,43]. Let $\{\lambda_1, \ldots \lambda_N\}$ be a solution of Bethe equations with $N < \frac{M}{2}$ and and $\{\mu_1, \ldots \mu_{N+\ell}\}$ a generic set of complex parameters with $\ell \leqslant M - 2N$. Then the following determinant formula holds

$$\langle\Psi(\{\mu\})|\Psi_\ell(\{\lambda\})\rangle = \frac{(-1)^{N\ell + \frac{\ell^2}{2}} \ell! \prod_{k=1}^{N+\ell} q(\mu_k - i)}{\prod_{j>k}^{N}(\lambda_j - \lambda_k) \prod_{j>k}^{N+\ell}(\mu_k - \mu_j)} \det_{N+\ell} \mathcal{M}^{(\ell)}(\{\lambda\}|\{\mu\}),$$

$$\mathcal{M}_{j,k}^{(\ell)}(\{\lambda\}|\{\mu\}) = \mathfrak{a}(\mu_k)t(\mu_k - \lambda_j) - t(\lambda_j - \mu_k), \qquad \text{for } j \leqslant N,$$

$$\mathcal{M}_{j,k}^{(\ell)}(\{\lambda\}|\{\mu\}) = \mathfrak{a}(\mu_k)(\mu_k + i)^{j-N-1} - \mu_k^{j-N-1}, \qquad \text{for } j > N. \tag{2.34}$$

Note that this last determinant formula for scalar products also appears in the framework of the separation of variables approach [44].

Using the $\mathfrak{su}(2)$ symmetry of the model, we can show that the following relations hold between the matrix elements of the local operators and norms of triplet states

$$\langle\Psi_1(\mu_{h_1}, \mu_{h_2})|\sigma_m^z|\Psi_g\rangle \quad = -2\langle\Psi_0(\mu_{h_1}, \mu_{h_2})|\sigma_m^+|\Psi_g\rangle, \tag{2.35}$$

$$\langle\Psi_g|\sigma_m^z|\Psi_1(\mu_{h_1}, \mu_{h_2})\rangle \quad = \quad \langle\Psi_g|\sigma_m^+|\Psi_2(\mu_{h_1}, \mu_{h_2})\rangle, \tag{2.36}$$

$$\langle\Psi_1(\mu_{h_1}, \mu_{h_2})|\Psi_1(\mu_{h_1}, \mu_{h_2})\rangle = 2\langle\Psi_0(\mu_{h_1}, \mu_{h_2})|\Psi_0(\mu_{h_1}, \mu_{h_2})\rangle. \tag{2.37}$$

It is worthwhile to note that these identities and the determinant formula (2.34) permits us to write the triplet form factor as simple determinants without any extra sums.

Now, using the solution of the quantum inverse problem (2.30) we obtain the determinant representation for the form factor

$$|\mathcal{F}_z(\mu_{h_1}, \mu_{h_2})|^2 = -2 \prod_{j=1}^{\frac{M}{2}-1} \frac{q_g(\mu_j - i)}{q_e(\mu_j - i)} \prod_{k=1}^{\frac{M}{2}} \frac{q_e(\lambda_k - i)}{q_g(\lambda_k - i)}$$

$$\times \frac{\det_{\frac{M}{2}} \mathcal{M}(\{\lambda\}|\{\mu_1 \dots \mu_{\frac{M}{2}-1}, \frac{i}{2}\}) \det_{\frac{M}{2}+1} \mathcal{M}^{(2)}(\{\mu\}|\{\lambda_1, \dots \lambda_{\frac{M}{2}}, \frac{i}{2}\})}{\det_{\frac{M}{2}} \mathcal{N}(\{\lambda\}) \det_{\frac{M}{2}-1} \mathcal{N}(\{\mu\})}. \tag{2.38}$$

This formula is valid for any triplet excited state and not just limited to the two-spinon sector and hence it can be used for any non-trivial form factor.

## 3 Computation of determinants

### 3.1 Integral equations

To compute the form factors using the determinant representation (2.38) we will study the action of the inverse Gaudin matrix on the Slavnov matrix. More precisely we want to compute the following matrices

$$F_g = \mathcal{N}^{-1}(\{\lambda\}) \mathcal{M}(\{\lambda\}|\{\mu_1 \dots \mu_{\frac{M}{2}-1}, \frac{i}{2}\}), \tag{3.1}$$

$$F_e = \mathcal{N}^{(2)^{-1}}(\{\mu\}) \mathcal{M}^{(2)}(\{\mu\}|\{\lambda_1, \dots \lambda_{\frac{M}{2}}, \frac{i}{2}\}), \tag{3.2}$$

where $\mathcal{N}^{(2)}$ is the $(\frac{M}{2}+1) \times (\frac{M}{2}+1)$ Gaudin matrix with two additional rows and columns

$$\mathcal{N}_{jk}^{(2)}(\{\mu\}) = \mathcal{N}_{jk}(\{\mu\}) \qquad \text{if} \quad j,k \leqslant \frac{M}{2} - 1,$$
$$\mathcal{N}_{jk}^{(2)}(\{\mu\}) = \delta_{jk} \qquad \text{if} \quad j \geqslant \frac{M}{2} \text{ or } k \geqslant \frac{M}{2}. \tag{3.3}$$

For the first matrix we obtain the following system of linear equations

$$\mathfrak{a}_g'(\lambda_j) F_{gj,k} - \sum_{a=1}^{\frac{M}{2}} K(\lambda_j - \lambda_a) F_{ga,k} = \mathfrak{a}_g(\mu_k) t(\mu_k - \lambda_j) - t(\lambda_j - \mu_k). \tag{3.4}$$

The approach we use here is very close to the one used in [45] and it is based on replacement of sums by contour integrals. It is evident that if a meromorphic function $G_g(\lambda; \mu_k)$ satisfies the following equation for any $\lambda$ in a strip around the real axis $-\frac{1}{2} < \mathfrak{J}(\lambda) < \frac{1}{2}$

$$G_g(\lambda; \mu_k) - \sum_{a=1}^{\frac{M}{2}} \text{Res} \left( K(\lambda - \nu) \frac{G_g(\nu; \mu_k)}{1 + \mathfrak{a}_g(\nu)} \right) \Bigg|_{\nu = \lambda_a} = \mathfrak{a}_g(\mu_k) t(\mu_k - \lambda) - t(\lambda - \mu_k), \tag{3.5}$$

it would give the unique solution of the system (3.4)

$$\mathfrak{a}_g'(\lambda_j) F_{gj,k} = G_g(\lambda_j; \mu_k). \tag{3.6}$$

From the right hand side of the equation (3.5) it is clear that the only singularity of $G_g(\lambda; \mu_k)$ in the strip $-\frac{1}{2} < \mathfrak{I}(\lambda) < \frac{1}{2}$ is the point $\lambda = \mu_k$. It is easy to compute its residue at this point:

$$\text{Res}\, G_g(\lambda; \mu_k)\big|_{\lambda=\mu_k} = -1 - \mathfrak{a}_g(\mu_k). \tag{3.7}$$

Taking into account this additional simple pole we can rewrite the sum in (3.5) as a contour integral if $\mu_k$ is real

$$G_g(\lambda; \mu_k) - \frac{1}{2\pi i} \oint_\Gamma d\nu\, K(\lambda - \nu) \frac{G_g(\nu; \mu_k)}{1 + \mathfrak{a}_g(\nu)} = \mathfrak{a}_g(\mu_k) t(\mu_k - \lambda) - t(\lambda - \mu_k) + K(\lambda - \mu_k), \tag{3.8}$$

where the contour $\Gamma$ includes all the Bethe roots for the ground state (zeroes of $1 + \mathfrak{a}_g(\nu)$) and the point $\mu_k$. It can be chosen as a rectangle with vertices at $-\Lambda - i\alpha$, $\Lambda - i\alpha$, $\Lambda + i\alpha$ and $-\Lambda + i\alpha$, with $\Lambda > \lambda_{\max}$ (the maximal Bethe root) and $\alpha < \frac{1}{2}$. Since $K(\lambda) = t(\lambda) + t(-\lambda)$, we can write the function $G_g(\lambda; \mu)$ as

$$G_g(\lambda; \mu) = \big(1 + \mathfrak{a}_g(\mu)\big)\rho_g(\lambda; \mu), \tag{3.9}$$

where $\rho_g(\lambda; \mu)$ solves the following integral equation

$$\rho_g(\lambda; \mu) - \frac{1}{2\pi i} \oint_\Gamma d\nu\, K(\lambda - \nu) \frac{\rho_g(\nu; \mu)}{1 + \mathfrak{a}_g(\nu)} = t(\mu - \lambda). \tag{3.10}$$

If $\mathfrak{R}(\lambda)$ is in the bulk (sufficiently far from the ends of distribution) the counting function $\mathfrak{a}_g(\lambda)$ has a following exponential behaviour for complex arguments in the thermodynamic limit : $\mathfrak{a}_g(\lambda) = O(M^\infty)$ if $\mathfrak{I}(\lambda) < 0$ and $\mathfrak{a}_g(\lambda) = O(M^{-\infty})$ if $\mathfrak{I}(\lambda) > 0$. The "bulk assumption" used throughout this paper, is a generalisation of the condensation property of Bethe roots (2.19,2.20) for functions with some singularities on the real axis. We will assume for all the functions appearing in this paper under integrals like in (3.10) that only this bulk behaviour is pertinent and the tails of distribution do not contribute to the leading order (for the ground and excited states)

$$\oint_\Gamma d\nu\, K(\lambda - \nu) \frac{\rho_g(\nu; \mu)}{1 + \mathfrak{a}_g(\nu)} = -\int_{\mathbb{R}+i\alpha} d\nu\, K(\lambda - \nu)\rho_g(\nu; \mu) + o\left(\frac{1}{M}\right). \tag{3.11}$$

We will also assume that the subdominant corrections $o\left(\frac{1}{M}\right)$ do not influence the leading order of the determinants.

We obtain finally a simple equation for the matrix elements

$$\rho_g(\lambda; \mu) + \frac{1}{2\pi i} \int_{\mathbb{R}+i\alpha} d\nu\, K(\lambda - \nu)\rho_g(\nu; \mu) = t(\mu - \lambda) + o\left(\frac{1}{M}\right). \tag{3.12}$$

It is easy to recognise here that the above equation satisfied by $\rho_g(\lambda; \mu)$ is the Lieb equation for the density of Bethe roots (2.16) and as a result, we obtain a very simple solution

$$\rho_g(\lambda; \mu) = \frac{\pi}{\sinh \pi(\mu - \lambda)} + o\left(\frac{1}{M}\right). \tag{3.13}$$

In a similar way we can study the last line of the matrix $F_g$. Since $\mathfrak{a}_g(\frac{i}{2}) = 0$, we obtain

$$G_g\left(\lambda; \frac{i}{2}\right) - \frac{1}{2\pi i} \oint_\Gamma d\nu\, K(\lambda - \nu) \frac{G_g\left(\nu; \frac{i}{2}\right)}{1 + \mathfrak{a}_g(\nu)} = -t\left(\lambda - \frac{i}{2}\right), \tag{3.14}$$

leading directly to the Lieb equation

$$G_g\left(\lambda;\frac{i}{2}\right) = -i\frac{\pi}{\cosh\pi\lambda}. \tag{3.15}$$

This result means that the first ratio of determinants is reduced to the Cauchy determinant of densities and it can be easily computed (we assume that corrections of order $o\left(\frac{1}{M}\right)$ do not contribute to the leading order of the determinant in the thermodynamic limit).

$$\det_{\frac{M}{2}} F_g = \pi^{\frac{M}{2}}\frac{\displaystyle\prod_{k=1}^{\frac{M}{2}-1}\left(1+\mathfrak{a}_g(\mu_k)\right)}{\displaystyle\prod_{j=1}^{\frac{M}{2}}\mathfrak{a}'(\lambda_j)}\frac{\displaystyle\prod_{j<k}^{\frac{M}{2}}\sinh\pi(\mu_j-\mu_k)\sinh\pi(\lambda_k-\lambda_j)}{\displaystyle\prod_{j=1}^{\frac{M}{2}}\prod_{k=1}^{\frac{M}{2}}\sinh\pi(\mu_k-\lambda_j)}. \tag{3.16}$$

where we set $\mu_{\frac{M}{2}} = \frac{i}{2}$. Note that this computation can be performed in a very similar way for the XXZ chain. For the excited states with complex roots the treatment of the integration contours should be slightly modified.

The second matrix $F_e$ can be computed following the same steps but turns out to be more complicated. The elements of first $\frac{M}{2}-1$ rows solve the following system of linear equations, while the last two rows are the same as in the Foda-Wheeler formula (2.34)

$$\mathfrak{a}'_e(\mu_j)F_{ej,k} - \sum_{a=1}^{\frac{M}{2}-1}K(\mu_j-\mu_a)F_{ea,k} = \mathfrak{a}_e(\lambda_k)t(\lambda_k-\mu_j)-t(\mu_j-\lambda_k) \quad \text{for } j\leqslant\frac{M}{2}-1,$$

$$F_{e\frac{M}{2},k} = \mathfrak{a}_e(\lambda_k)-1, \qquad F_{e\frac{M}{2}+1,k} = \mathfrak{a}_e(\lambda_k)(\lambda_k+i)-\lambda_k. \tag{3.17}$$

This system can be written in a residue form for a meromorphic function $G_e(\mu,\lambda)$

$$G_e(\mu;\lambda_k) - \sum_{a=1}^{\frac{M}{2}-1}\text{Res}\left(K(\mu-\nu)\frac{G_e(\nu;\lambda_k)}{1+\mathfrak{a}_e(\nu)}\right)\Bigg|_{\nu=\mu_a} = \mathfrak{a}_e(\lambda_k)t(\lambda_k-\mu)-t(\mu-\lambda_k), \tag{3.18}$$

and it gives the unique solution of the system (3.17)

$$\mathfrak{a}'_e(\mu_j)F_{ej,k} = G_e(\mu_j;\lambda_k). \tag{3.19}$$

There is once again an extra simple real pole of $G_e(\mu,\lambda_k)$ at the point $\mu = \lambda_k$. However for the excited states there are also extra real poles at the points corresponding to holes $\mu = \mu_{h_a}$. Hence the integral equation can be written as follows.

$$G_e(\mu;\lambda_k) - \frac{1}{2\pi i}\oint_\Gamma d\nu\, K(\mu-\nu)\frac{G_e(\nu;\lambda_k)}{1+\mathfrak{a}_e(\nu)} =$$

$$(\mathfrak{a}_e(\lambda_k)+1)t(\lambda_k-\mu) - \sum_{a=1}^{n_h}K(\mu-\mu_{h_a})\frac{G_e(\mu_{h_a};\lambda_k)}{\mathfrak{a}'_e(\mu_{h_a})}. \tag{3.20}$$

Following the same procedure with the integration contour using the properties of the counting function we obtain the following solution

$$G_e(\mu;\lambda_k) = (\mathfrak{a}_e(\lambda_k)+1)\rho_g(\mu,\lambda_k) - 2\pi i\sum_{a=1}^{n_h}\rho_h(\mu-\mu_{h_a})\frac{G_e(\mu_{h_a};\lambda_k)}{\mathfrak{a}'_e(\mu_{h_a})} + o\left(\frac{1}{M}\right), \tag{3.21}$$

where $\rho_g(\mu, \lambda)$ solves (3.12) and $\rho_h(\mu)$ is given by (2.21). It remains to determine the values of $G_e(\mu_{h_a}; \lambda_k)$. Setting $\mu = \mu_{h_a}$ in (3.21) we obtain a system of $n_h$ linear equations,

$$\sum_{b=1}^{n_h} \mathcal{H}_{ab} G_e(\mu_{h_b}; \lambda_k) = (\mathfrak{a}_e(\lambda_k) + 1) \rho_g(\mu_{h_a}, \lambda_k), \tag{3.22}$$

with a matrix $\mathcal{H}$

$$\mathcal{H}_{ab} = \delta_{ab} + 2\pi i \frac{\rho_h(\mu_{h_a} - \mu_{h_b})}{\mathfrak{a}'_e(\mu_{h_b})}. \tag{3.23}$$

Note that this $n_h \times n_h$ matrix is completely defined by the positions of holes. For holes in the bulk, we have

$$\frac{1}{\mathfrak{a}'_e(\mu_{h_a})} = O\left(\frac{1}{M}\right),$$

theorefore for the leading order of $G_e(\mu_{h_a}; \lambda_k)$ we obtain

$$G_e(\mu_{h_a}; \lambda_k) = (\mathfrak{a}_e(\lambda_k) + 1) \rho_g(\mu_{h_a}, \lambda_k) + O\left(\frac{1}{M}\right). \tag{3.24}$$

As the additional terms in (3.21) are already of order $O(\frac{1}{M})$ the corrections can be neglected and we get the following expression for the matrix elements with $j \leqslant \frac{M}{2} - 1$ and $k = 1, \ldots, \frac{M}{2} + 1$,

$$F_{e\,j,k} = \frac{\mathfrak{a}_e(\lambda_k) + 1}{\mathfrak{a}'_e(\mu_j)} \left( \frac{\pi}{\sinh \pi(\lambda_k - \mu_j)} \right.$$

$$\left. - 2\pi i \sum_{a=1}^{n_h} \frac{\rho_h(\mu_j - \mu_{h_a})}{\mathfrak{a}'_e(\mu_{h_a})} \frac{\pi}{\sinh \pi(\lambda_k - \mu_{h_a})} + o\left(\frac{1}{M}\right) \right), \quad (3.25)$$

we set here again $\lambda_{\frac{M}{2}+1} = \frac{i}{2}$. It is once again a Cauchy matrix but this time with addition of a matrix of rank $n_h$. There are also two additional Foda-Wheeler rows. This structure can be observed for triplets with any number of spinons. However the complex roots need a special treatment and will be considered in forthcoming publications. In this paper we limit our analysis to the case of $n_h = 2$.

Taking into account all the factors and using (2.10) we obtain the following expression for the form factors in terms of Cauchy determinants

$$|\mathcal{F}_z(\mu_{h_1}, \mu_{h_2})|^2 = -2 \frac{\prod_{j=1}^{\frac{M}{2}} \chi(\lambda_j)}{\prod_{k=1}^{\frac{M}{2}-1} \chi(\mu_k)} \frac{\prod_{j=1}^{\frac{M}{2}} \prod_{k=1}^{\frac{M}{2}-1} (\lambda_j - \mu_k)^2}{\prod_{j\neq k}^{\frac{M}{2}} (\lambda_j - \lambda_k) \prod_{j\neq k}^{\frac{M}{2}-1} (\mu_j - \mu_k)} \det_{\frac{M}{2}} \mathcal{R}_g \det_{\frac{M}{2}+1} \mathcal{R}_e, \tag{3.26}$$

where $\mathcal{R}_g$ and $\mathcal{R}_e$ are corresponding Cauchy and modified Cauchy matrix.

$$\mathcal{R}_{g\,j,k} = \frac{\pi}{\sinh \pi(\mu_k - \lambda_j)}, \tag{3.27}$$

$$\mathcal{R}_{e\,j,k} = \frac{\pi}{\sinh \pi(\lambda_k - \mu_j)} - 2\pi i \sum_{a=1}^{2} \frac{\rho_h(\mu_j - \mu_{h_a})}{\mathfrak{a}'_e(\mu_{h_a})} \frac{\pi}{\sinh \pi(\lambda_k - \mu_{h_a})}, \quad j \leqslant \frac{M}{2} - 1, \tag{3.28}$$

$$\mathcal{R}_{e\,\frac{M}{2},k} = \frac{\mathfrak{a}_e(\lambda_k) - 1}{\mathfrak{a}_e(\lambda_k) + 1}, \qquad \mathcal{R}_{e\,\frac{M}{2}+1,k} = \frac{\mathfrak{a}_e(\lambda_k)(\lambda_k + i) - \lambda_k}{\mathfrak{a}_e(\lambda_k) + 1}. \tag{3.29}$$

Once again, such representation for the form factor is rather general and can be used (with slight modifications due to the presence of complex roots) for any excited state. This formula can be considered as one of the most important intermediate result of the paper. There is no Fredholm determinant, we obtain essentially two Cauchy determinants with a finite rank additional matrix.

### 3.2 Cauchy determinant extraction

While the determinant of the Cauchy matrix $\mathcal{R}_g$ can be directly computed (3.16) the second matrix $\mathcal{R}_e$ contains two Foda-Wheeler rows as well as a rank 2 (in the two-spinon case) additional matrix.

To compute this determinant we first introduce the following $(\frac{M}{2} + 1) \times (\frac{M}{2} + 1)$ Cauchy matrix

$$\mathcal{C}_{jk} = \frac{\pi}{\sinh \pi (\lambda_k - \nu_j)}, \quad j, k = 1, \ldots, \frac{M}{2} + 1, \tag{3.30}$$

where we set as usual $\lambda_{\frac{M}{2}+1} = \frac{i}{2}$ and parameters $\nu_k$ is defined as follows

$$\nu_j = \mu_j \quad \text{for} \quad j \leqslant \frac{M}{2} - 1, \qquad \nu_{\frac{M}{2}} = \mu_{h_1}, \qquad \nu_{\frac{M}{2}+1} = \mu_{h_2}.$$

The determinant of the matrix $\mathcal{R}_e$ can be written as

$$\det \mathcal{R}_e = \det \mathcal{C} \det(\mathcal{R}_e \mathcal{C}^{-1}).$$

The determinant and inverse of the Cauchy matrix $\mathcal{C}$ can be easily computed. We denote the residual matrix by $\mathcal{P} = \mathcal{R}_e \mathcal{C}^{-1}$. It can be easily shown that it contains an identity block

$$\mathcal{P}_{j,k} = \delta_{jk}, \quad j, k \leqslant \frac{M}{2} - 1, \tag{3.31}$$

and a block two columns of size $\frac{M}{2} - 1$

$$\mathcal{P}_{j,k} = -2\pi i \frac{\rho_h(\nu_j - \nu_k)}{\mathfrak{a}'_e(\nu_k)}, \quad k = \frac{M}{2}, \frac{M}{2} + 1, \quad j \leqslant \frac{M}{2} - 1. \tag{3.32}$$

This computation is straightforward and immediately follows from the Cauchy structure of the first $\frac{M}{2} - 1$ rows and the additional rank 2 matrix.

The most complicated part is the action of the inverse Cauchy on the Foda-Wheeler rows,

$$\mathcal{P}_{\frac{M}{2},k} = \frac{1}{\pi} \frac{\prod\limits_{b=1}^{\frac{M}{2}+1} \sinh \pi(\nu_k - \lambda_b)}{\prod\limits_{b \neq k} \sinh \pi(\nu_k - \nu_b)}$$

$$\times \sum_{a=1}^{\frac{M}{2}+1} \frac{\prod\limits_{b=1}^{N+1} \sinh \pi(\lambda_a - \nu_b)}{\prod\limits_{b \neq a} \sinh \pi(\lambda_a - \lambda_b)} \frac{1}{\sinh \pi(\lambda_a - \nu_k)} \frac{\mathfrak{a}_e(\lambda_a) - 1}{\mathfrak{a}_e(\lambda_a) + 1}, \tag{3.33}$$

$$\mathcal{P}_{\frac{M}{2}+1,k} = \frac{1}{\pi} \frac{\prod\limits_{b=1}^{\frac{M}{2}+1} \sinh \pi(\nu_k - \lambda_b)}{\prod\limits_{b \neq k} \sinh \pi(\nu_k - \nu_b)}$$

$$\times \sum_{a=1}^{N+1} \frac{\prod\limits_{b=1}^{\frac{M}{2}+1} \sinh \pi(\lambda_a - \nu_b)}{\prod\limits_{b \neq a} \sinh \pi(\lambda_a - \lambda_b)} \frac{1}{\sinh \pi(\lambda_a - \nu_k)} \frac{\mathfrak{a}_e(\lambda_a)(\lambda_a + i) - \lambda_a}{\mathfrak{a}_e(\lambda_a) + 1}. \quad (3.34)$$

We introduce the following product

$$\Phi(\lambda) = \prod_{b=1}^{\frac{M}{2}+1} \frac{\sinh \pi(\lambda - \nu_b)}{\sinh \pi(\lambda - \lambda_b)}. \quad (3.35)$$

This meromorphic function has following essential properties

- Periodicity $\Phi(\lambda + i) = \Phi(\lambda)$,

- Simple poles at the points $\lambda = \lambda_a + in$, $n \in \mathbb{Z}$, and

- Simple zeros $\lambda = \nu_a + in$, $n \in \mathbb{Z}$.

Using these properties we can rewrite sums in (3.33,3.34) as contour integrals

$$\mathcal{P}_{\frac{M}{2},k} = \frac{1}{2\pi i \, \Phi'(\nu_k)} \oint_\Gamma \frac{d\lambda}{\sinh \pi(\lambda - \nu_k)} \Phi(\lambda) \frac{\mathfrak{a}_e(\lambda) - 1}{\mathfrak{a}_e(\lambda) + 1} + \frac{2}{\mathfrak{a}'_e(\nu_k)}, \quad (3.36)$$

$$\mathcal{P}_{\frac{M}{2}+1,k} = \frac{1}{2\pi i \, \Phi'(\nu_k)} \oint_\Gamma \frac{d\lambda}{\sinh \pi(\lambda - \nu_k)} \Phi(\lambda) \frac{\mathfrak{a}_e(\lambda)(\lambda + i) - \lambda}{\mathfrak{a}_e(\lambda) + 1} + \frac{2\nu_k + i}{\mathfrak{a}'_e(\nu_k)}. \quad (3.37)$$

Here it is essential to note that all zeroes of $\mathfrak{a}_e(\lambda) + 1$ except $\lambda = \nu_k$ produce a pole with a zero residue since the function $\Phi(\lambda)$ has zeroes at the same points. The residues of the only remaining additional pole $\lambda = \nu_k$, give rise to the additional terms appearing in both these expressions. The contour $\Gamma$ is chosen in such a way that it encircles all the poles $\lambda_j$ but not the poles $\lambda_j + in$ with $n \neq 0$. In particular as the integrand decreases exponentially at infinity we can set as two horizontal lines

$$\oint_\Gamma = \int_{\mathbb{R} - i\epsilon} - \int_{\mathbb{R} + i/2 + i\epsilon}, \quad (3.38)$$

with $\epsilon < 1/2$.

Taking into account that $\mathfrak{a}(\lambda) \to \infty$ if $\mathfrak{I}(\lambda) < 0$ and $\mathfrak{a}(\lambda) \to 0$ if $\mathfrak{I}(\lambda) > 0$ (we apply again the bulk assumption here) and using the periodicity of the function $\Phi(\lambda)$ we obtain in the thermodynamic limit

$$\oint_\Gamma \frac{\Phi(\lambda) \, d\lambda}{\sinh \pi(\lambda - \nu_k)} \frac{\mathfrak{a}_e(\lambda) - 1}{\mathfrak{a}_e(\lambda) + 1} = \int_{\mathbb{R} - i\epsilon} \frac{\Phi(\lambda) \, d\lambda}{\sinh \pi(\lambda - \nu_k)} + \int_{\mathbb{R} + i/2 + i\epsilon} \frac{\Phi(\lambda) \, d\lambda}{\sinh \pi(\lambda - \nu_k)}$$

$$= -\int_{\mathbb{R} + i - i\epsilon} \frac{\Phi(\lambda) \, d\lambda}{\sinh \pi(\lambda - \nu_k)} + \int_{\mathbb{R} + i/2 + i\epsilon} \frac{\Phi(\lambda) \, d\lambda}{\sinh \pi(\lambda - \nu_k)}, \quad (3.39)$$

$$\oint_{\Gamma} \frac{\Phi(\lambda)\,d\lambda}{\sinh\pi(\lambda-\nu_k)} \frac{\mathfrak{a}_e(\lambda)(\lambda+i)-\lambda}{\mathfrak{a}_e(\lambda)+1} = \int_{\mathbb{R}-i\epsilon} \frac{(\lambda+i)\Phi(\lambda)\,d\lambda}{\sinh\pi(\lambda-\nu_k)} + \int_{\mathbb{R}+i/2+i\epsilon} \frac{\lambda\,\Phi(\lambda)\,d\lambda}{\sinh\pi(\lambda-\nu_k)}$$

$$= -\int_{\mathbb{R}+i-i\epsilon} \frac{\lambda\,\Phi(\lambda)\,d\lambda}{\sinh\pi(\lambda-\nu_k)} + \int_{\mathbb{R}+i/2+i\epsilon} \frac{\lambda\,\Phi(\lambda)\,d\lambda}{\sinh\pi(\lambda-\nu_k)}. \quad (3.40)$$

In both cases we get integrals over a closed contour of a meromorphic function without any poles inside, so both integrals are vanishing and we obtain our final result for the last 2 rows

$$\mathcal{P}_{\frac{M}{2},k} = \frac{2}{\mathfrak{a}'_e(\nu_k)}, \qquad \mathcal{P}_{\frac{M}{2}+1,k} = \frac{2\nu_k+i}{\mathfrak{a}'_e(\nu_k)}. \quad (3.41)$$

It remains to compute the determinant of the block matrix $\mathcal{P}$ which can be reduced to a determinant of a $2 \times 2$ matrix

$$\mathcal{P} = \begin{pmatrix} \mathcal{I} & \mathcal{B} \\ \mathcal{A} & \mathcal{D} \end{pmatrix}, \qquad \det_{N+1}\mathcal{P} = \det_2(\mathcal{D}-\mathcal{A}\mathcal{B}), \quad (3.42)$$

where $\mathcal{I}$ is $(\frac{M}{2}-1) \times (\frac{M}{2}-1)$ identity matrix, $\mathcal{B}$ is a $2 \times (\frac{M}{2}-1)$ matrix etc.

Our last step is to compute the matrix product $\mathcal{A}\mathcal{B}$. These simple sums can be easily computed using the condensation property of the Bethe roots (2.20)

$$\sum_{k=1}^{\frac{M}{2}-1} \mathcal{A}_{1k}\mathcal{B}_{ka} = \frac{2}{\mathfrak{a}'_e(\mu_{h_a})} \left( \int_{-\infty}^{\infty} \rho_h(\mu-\mu_{h_a})\,d\mu + O\left(\frac{1}{M}\right) \right), \quad (3.43)$$

$$\sum_{k=1}^{\frac{M}{2}-1} \mathcal{A}_{2k}\mathcal{B}_{ka} = \frac{2}{\mathfrak{a}'_e(\mu_{h_a})} \left( \int_{-\infty}^{\infty} \left(\mu+\frac{i}{2}\right)\rho_h(\mu-\mu_{h_a})\,d\mu + O\left(\frac{1}{M}\right) \right). \quad (3.44)$$

With the following simple observation

$$\int_{-\infty}^{\infty} \rho_h(\lambda)\,d\lambda = \frac{1}{2} \quad (3.45)$$

and the fact that $\rho_h(\lambda)$ is an even function we easily obtain

$$\mathcal{D}-\mathcal{A}\mathcal{B} = \begin{pmatrix} \frac{1}{\mathfrak{a}'_e(\mu_{h_1})} & \frac{1}{\mathfrak{a}'_e(\mu_{h_2})} \\ \frac{\mu_{h_1}+\frac{i}{2}}{\mathfrak{a}'_e(\mu_{h_1})} & \frac{\mu_{h_2}+\frac{i}{2}}{\mathfrak{a}'_e(\mu_{h_2})} \end{pmatrix}, \quad (3.46)$$

and

$$\det_{\frac{M}{2}+1}\mathcal{P} = \frac{1}{\mathfrak{a}'_e(\mu_{h_1})\,\mathfrak{a}'_e(\mu_{h_2})}(\mu_{h_2}-\mu_{h_1}). \quad (3.47)$$

Finally we note that in the thermodynamic limit, $\mathfrak{a}'_e(\mu_{h_a})$ can be expanded as

$$\mathfrak{a}'_e(\mu_{h_a}) = 2\pi i \left( M\rho_g(\mu_{h_a}) + \sum_{b=1}^{2} \rho_h(\mu_{h_a}-\mu_{h_b}) \right) + o(1), \quad 1 \le a \le n_h. \quad (3.48)$$

Assuming the hole rapidities $\mu_{h_a}$ lie in the bulk of the distribution of the Bethe roots, it can be readily seen that the thermodynamic limit of these terms is dominated by ground state root-density function with a coefficient which scales with $M$. This gives

$$\det \mathcal{P} = \frac{\mu_{h_1} - \mu_{h_2}}{\pi^2 M^2} \prod_{a=1}^{2} \cosh \pi \mu_{h_a} + o\left(\frac{1}{M^2}\right). \tag{3.49}$$

Note that this determinant gives the expected scaling $M^{-2}$ for the form factor. In the next subsection we will compute the scaled form factor in the thermodynamic limit

$$|\widehat{\mathcal{F}_z}|^2 = \lim_{M \to \infty} M^2 |\mathcal{F}_z|^2. \tag{3.50}$$

We would like to mention that this integer scaling is typical for the zero magnetic field form factors with holes in the bulk. It is easy to check that in this case the boundary values of the shift functions responsible for the non-integer scaling in the finite magnetic field case [25] vanish and only the integer term remains.

## 3.3 Thermodynamic limit

In this subsection, we will compute the thermodynamic limit of the form-factor (3.26) $|\mathcal{F}_z|^2$ Now all the determinants in the representation (3.26) can be computed. The determinant of the Cauchy matrix $\mathcal{C}$ (3.30) can be written as

$$\det \mathcal{C} = \pi^{\frac{M}{2}+1} \frac{\sinh \pi(\mu_{h_2} - \mu_{h_1})}{\cosh \pi \mu_{h_1} \cosh \pi \mu_{h_2}} \frac{\displaystyle\prod_{j>k}^{\frac{M}{2}} \sinh \pi(\lambda_j - \lambda_k) \prod_{j>k}^{\frac{M}{2}-1} \sinh \pi(\mu_k - \mu_j)}{\displaystyle\prod_{j=1}^{\frac{M}{2}} \prod_{k=1}^{\frac{M}{2}-1} \sinh \pi(\lambda_j - \mu_k)}$$
$$\times \prod_{a=1}^{2} \frac{\displaystyle\prod_{j=1}^{\frac{M}{2}-1} \sinh \pi(\mu_{h_a} - \mu_j)}{\displaystyle\prod_{j=1}^{\frac{M}{2}} \sinh \pi(\mu_{h_a} - \lambda_j)} \frac{\displaystyle\prod_{j=1}^{\frac{M}{2}} \sinh \pi(\frac{i}{2} - \lambda_j)}{\displaystyle\prod_{j=1}^{\frac{M}{2}-1} \sinh \pi(\frac{i}{2} - \mu_j)}, \tag{3.51}$$

and similarly the determinant of the Cauchy matrix $\mathcal{R}_g$ can be written as

$$\det \mathcal{R}_g = \pi^{\frac{M}{2}} \frac{\displaystyle\prod_{j>k}^{\frac{M}{2}-1} \sinh \pi(\mu_j - \mu_k) \prod_{j>k}^{\frac{M}{2}} \sinh \pi(\lambda_k - \lambda_j)}{\displaystyle\prod_{j=1}^{\frac{M}{2}-1} \prod_{k=1}^{\frac{M}{2}} \sinh \pi(\mu_j - \lambda_k)} \cdot \frac{\displaystyle\prod_{j=1}^{\frac{M}{2}-1} \sinh \pi(\frac{i}{2} - \mu_j)}{\displaystyle\prod_{j=1}^{\frac{M}{2}} \sinh \pi(\frac{i}{2} - \lambda_j)}. \tag{3.52}$$

Here we can easily see that the terms explicitly containing the extra rapidities $\mu_{\frac{M}{2}} = \frac{i}{2}$ and $\lambda_{\frac{M}{2}+1} = \frac{i}{2}$ can be cancelled out in the the product of two Cauchy determinants which together

with (3.49) gives us the following expression for the form-factors (3.26)

$$
|\mathcal{F}_z|^2 = 2\pi^{M-1} \frac{\mu_{h_1} - \mu_{h_2}}{M^2} \sinh \pi(\mu_{h_1} - \mu_{h_2}) \frac{\prod\limits_{j=1}^{\frac{M}{2}} \chi(\lambda_j)}{\prod\limits_{j=1}^{\frac{M}{2}-1} \chi(\mu_j)} \prod_{a=1}^{2} \frac{\prod\limits_{j=1}^{\frac{M}{2}-1} \sinh \pi(\mu_{h_a} - \mu_j)}{\prod\limits_{j=1}^{\frac{M}{2}} \sinh \pi(\mu_{h_a} - \lambda_j)}
$$

$$
\times \frac{\prod\limits_{j=1}^{\frac{M}{2}} \prod\limits_{k=1}^{\frac{M}{2}-1} (\lambda_j - \mu_k)^2}{\prod\limits_{j\neq k}^{\frac{M}{2}} (\lambda_j - \lambda_k) \prod\limits_{j\neq k}^{\frac{M}{2}-1} (\mu_j - \mu_k)} \frac{\prod\limits_{j\neq k}^{\frac{M}{2}} \sinh \pi(\lambda_j - \lambda_k) \prod\limits_{j\neq k}^{\frac{M}{2}-1} \sinh \pi(\mu_j - \mu_k)}{\prod\limits_{j=1}^{\frac{M}{2}} \prod\limits_{k=1}^{\frac{M}{2}-1} \sinh^2 \pi(\lambda_j - \mu_k)}. \quad (3.53)
$$

The form factor is reduced now to double products over rapidities. Obviously these double products could be written in a logarithmic form replacing sums by integrals with densities. Here we suggest a more efficient way to compute these products in the thermodynamic limit based on infinite product factorisation.

The main idea behind the following computation of the Cauchy determinants can be described as follows: we use the Weierstrass infinite product formulas for the hyperbolic function and then we apply the auxiliary functions $\phi(\lambda)$ and $\chi(\lambda)$ whose thermodynamic limits are computed in the Appendix B to obtain a new infinite product from which we deduce our final result.

In the equation (3.53), the product can be factored as follows

$$
\frac{\prod\limits_{j\neq k}^{\frac{M}{2}} \sinh \pi(\lambda_j - \lambda_k) \prod\limits_{j\neq k}^{\frac{M}{2}-1} \sinh \pi(\mu_j - \mu_k)}{\prod\limits_{j=1}^{\frac{M}{2}} \prod\limits_{k=1}^{\frac{M}{2}-1} \sinh^2 \pi(\lambda_j - \mu_k)} = \pi^{-M+3} \frac{\prod\limits_{j\neq k}^{\frac{M}{2}} (\lambda_j - \lambda_k) \prod\limits_{j\neq k}^{\frac{M}{2}-1} (\mu_j - \mu_k)}{\prod\limits_{j=1}^{\frac{M}{2}} \prod\limits_{k=1}^{\frac{M}{2}-1} (\lambda_j - \mu_k)^2}
$$

$$
\times \prod_{j=1}^{\frac{M}{2}-1} \left| \frac{\prod\limits_{k=1}^{\frac{M}{2}} \Gamma(1 + \frac{\mu_j - \lambda_k}{2i}) \Gamma(\frac{1}{2} + \frac{\mu_j - \lambda_k}{2i})}{\prod\limits_{k=1}^{\frac{M}{2}-1} \Gamma(1 + \frac{\mu_j - \mu_k}{2i}) \Gamma(\frac{1}{2} + \frac{\mu_j - \mu_k}{2i})} \right|^2 \prod_{j=1}^{\frac{M}{2}} \left| \frac{\prod\limits_{k=1}^{\frac{M}{2}-1} \Gamma(1 + \frac{\lambda_j - \mu_k}{2i}) \Gamma(\frac{1}{2} + \frac{\lambda_j - \mu_k}{2i})}{\prod\limits_{k=1}^{\frac{M}{2}} \Gamma(1 + \frac{\lambda_j - \lambda_k}{2i}) \Gamma(\frac{1}{2} + \frac{\lambda_j - \lambda_k}{2i})} \right|^2, \quad (3.54)
$$

and

$$
\prod_{a=1}^{2} \frac{\prod\limits_{j=1}^{\frac{M}{2}-1} \sinh \pi(\mu_{h_a} - \mu_j)}{\prod\limits_{j=1}^{\frac{M}{2}} \sinh \pi(\mu_{h_a} - \lambda_j)} = \frac{1}{4\pi^4} \frac{\prod\limits_{j=1}^{\frac{M}{2}-1} \chi(\mu_j)}{\prod\limits_{j=1}^{\frac{M}{2}} \chi(\lambda_j)} \prod_{a=1}^{2} \left| \frac{\prod\limits_{j=1}^{\frac{M}{2}} \Gamma\left(\frac{1}{2} + \frac{(\mu_{h_a} - \lambda_j)}{2i}\right)}{\prod\limits_{j=1}^{\frac{M}{2}-1} \Gamma\left(\frac{1}{2} + \frac{(\mu_{h_a} - \mu_j)}{2i}\right)} \right|^4, \quad (3.55)
$$

where $\chi(\lambda)$ is defined in (2.24). Here we also define the function $\Omega(\lambda)$ for any real rapidity $\lambda$ as

$$
\Omega(\lambda) = \prod_{a=1}^{2} \frac{\Gamma^3\left(\frac{1}{2}\right)}{\left| \Gamma\left(\frac{1}{2} + \frac{\lambda - \mu_{h_a}}{2i}\right) \right|^4} \frac{\prod\limits_{j=1}^{\frac{M}{2}} \left| \Gamma\left(1 + \frac{\lambda - \lambda_j}{2i}\right) \right|^2 \left| \Gamma\left(\frac{1}{2} + \frac{\lambda - \lambda_j}{2i}\right) \right|^2}{\prod\limits_{j=1}^{\frac{M}{2}-1} \left| \Gamma\left(1 + \frac{\lambda - \mu_j}{2i}\right) \right|^2 \left| \Gamma\left(\frac{1}{2} + \frac{\lambda - \mu_j}{2i}\right) \right|^2}, \quad \lambda \in \mathbb{R}. \quad (3.56)
$$

In terms of this function we obtain the following representation for the form factor

$$|\mathcal{F}_z|^2 = \frac{\pi}{2M^2}(\mu_{h_1} - \mu_{h_2})\sinh\pi(\mu_{h_1} - \mu_{h_2})\frac{\prod_{j=1}^{\frac{M}{2}-1}\Omega(\mu_j)}{\prod_{j=1}^{\frac{M}{2}}\Omega(\lambda_j)}. \tag{3.57}$$

Using the Weierstrass form (A.1) of the $\Gamma$ function, we can express $\Omega$ as an infinite product

$$\Omega(\lambda) = \prod_{n=1}^{\infty}\Omega_n(\lambda), \tag{3.58}$$

where

$$\Omega_n(\lambda) = \frac{16n^2}{(n-\frac{1}{2})^6}|\phi(\lambda+2ni)|^2|\phi(\lambda+(2n-1)i)|^2\prod_{a=1}^{2}\left\{\left(n-\frac{1}{2}\right)^2 + \frac{(\lambda-\mu_{h_a})^2}{4}\right\}^2. \tag{3.59}$$

Here $\phi$ denotes the ratio of Baxter polynomials. Since the infinite product over $n$ is absolutely convergent we can compute the function $\Omega_n$ in the thermodynamic limit. Using (2.25) it is easy to see that $\phi(\lambda)$ satisfies

$$\phi(\lambda\pm 2ni)\phi(\lambda\pm(2n-1)i) = -\frac{1}{4}\prod_{a=1}^{2}\frac{1}{n-\frac{1}{2}\pm\frac{\lambda-\mu_{h_a}}{2i}}, \tag{3.60}$$

which gives

$$\Omega_n(\lambda) = \frac{n^2\prod_{a=1}^{2}\left\{\left(n-\frac{1}{2}\right)^2 + \frac{(\lambda-\mu_{h_a})^2}{4}\right\}}{(n-\frac{1}{2})^6}. \tag{3.61}$$

The remaining product over Bethe roots can be again rewritten in terms of the ratio of Baxter polynomials as follows

$$\frac{\prod_{j=1}^{\frac{M}{2}-1}\Omega_n(\mu_j)}{\prod_{j=1}^{\frac{M}{2}}\Omega_n(\lambda_j)} = \frac{(n-\frac{1}{2})^6}{16n^2}\prod_{a=1}^{2}\left|\phi(\mu_{h_a}+(2n-1)i)\right|^2. \tag{3.62}$$

Therefore the infinite product over $n$ can be written as follows

$$\prod_{n=1}^{\infty}\frac{\prod_{j=1}^{\frac{M}{2}-1}\Omega_n(\mu_j)}{\prod_{j=1}^{\frac{M}{2}}\Omega_n(\lambda_j)} = \prod_{n=1}^{\infty}\left\{\left(\frac{n-\frac{1}{2}}{n}\right)^2\left(\frac{\Gamma\left(n+\frac{1}{2}\right)}{\Gamma(n)}\right)^4\left|\frac{\Gamma(n-\frac{1}{2}+\frac{\mu_{h_1}-\mu_{h_2}}{2i})}{\Gamma(n+\frac{\mu_{h_1}-\mu_{h_2}}{2i})}\right|^4\right\}. \tag{3.63}$$

It can be readily seen using the Stirling's approximation for large $n$ that this infinite product is absolutely convergent. It can be written as a product of Barnes $G$ function using its Weierstrass form (A.6),

$$\prod_{n=1}^{\infty}\frac{\prod_{j=1}^{\frac{M}{2}-1}\Omega_n(\mu_j)}{\prod_{j=1}^{\frac{M}{2}}\Omega_n(\lambda_j)} = \frac{G^2\left(\frac{1}{2}\right)G^2(2)}{G^6\left(\frac{3}{2}\right)}\frac{G^2\left(1+\frac{\mu_{h_1}-\mu_{h_2}}{2i}\right)G^2\left(1-\frac{\mu_{h_1}-\mu_{h_2}}{2i}\right)}{G^2\left(\frac{1}{2}+\frac{\mu_{h_1}-\mu_{h_2}}{2i}\right)G^2\left(\frac{1}{2}-\frac{\mu_{h_1}-\mu_{h_2}}{2i}\right)}. \tag{3.64}$$

In the appendix A we give some important properties of the Barnes $G$ function, making this computation straightforward.

Therefore from (3.49), (3.50) and (3.64), we obtain the following thermodynamic limit for the form-factor

$$|\widehat{\mathcal{F}}_z|^2 = \frac{2}{G^4\left(\frac{1}{2}\right)} \left| \frac{G\left(\frac{\mu_{h_1}-\mu_{h_2}}{2i}\right) G\left(1 + \frac{\mu_{h_1}-\mu_{h_2}}{2i}\right)}{G\left(\frac{1}{2} + \frac{\mu_{h_1}-\mu_{h_2}}{2i}\right) G\left(\frac{3}{2} + \frac{\mu_{h_1}-\mu_{h_2}}{2i}\right)} \right|^2. \tag{3.65}$$

This formula is the final result of this paper, it remains to show that it is equivalent to the representation obtained in [16, 17] from the q-vertex operator approach. It can be easily checked using the integral representation for the $\log G$ function (A.7) that

$$|\widehat{\mathcal{F}}_z|^2 = 2e^{-I(\mu_{h_1}-\mu_{h_2})}, \quad I(\mu_{h_1}-\mu_{h_2}) = \int_0^\infty \frac{dt}{t} e^t \frac{\cos\left(2(\mu_{h_1}-\mu_{h_2})t\right)\cosh(2t)-1}{\cosh(t)\sinh(2t)}. \tag{3.66}$$

# 4 Conclusion

In this paper we compute the two-spinon form factor for the XXX spin chain in the algebraic Bethe ansatz framework. This result is the first step toward systematic computation of the form factors in the basis of Bethe vectors taking into account all the holes and complex roots. We show in particular that the ratio of the Slavnov determinant for the scalar products and Gaudin determinant for the norms can be written as a Cauchy determinant in the thermodynamic limit. This property is valid not only for the XXX chain but for both regimes of the XXZ chain in zero magnetic field. It means that this approach is directly generalisable to the anisotropic case. We believe that the method presented in this paper can lead to manageable expressions for the form factor for any state with a given configuration of holes and for a given solution of the higher level Bethe equations [40]. It would be also interesting to compare such results with the BJMST fermionic approach [29].

# Acknowledgements

N.K. is supported by CNRS, G.K. is supported by Carnot-Pasteur doctoral school (UBFC). This work has been supported by the EUR EIPHI program. N.K. would like to thank K.K. Kozlowski, J.M. Maillet, and V. Terras for numerous discussion. The authors would also like to thank Les Houches School of Physics where a part of this work was completed.

# A  Barnes $G$ function

In this appendix we remind the definition of the Barnes $G$ function or double $\Gamma$ function. It is a generalisation of the Euler's $\Gamma$ function, which satisfies $\Gamma(z+1) = z\Gamma(z)$, $\Gamma(1) = 1$ and admits the Weierstrass infinite product form

$$\Gamma(z) = \frac{e^{-\gamma z}}{z} \prod_{n=1}^\infty \frac{ne^{\frac{z}{n}}}{z+n}. \tag{A.1}$$

The logarithm of the $\Gamma$ function for $\Re(z) > 0$ admits the integral representation

$$\log \Gamma(z) = \int_0^\infty \frac{dt}{t} \left\{ (z-1)e^{-t} + \frac{e^{-zt} - e^{-t}}{1 - e^{-t}} \right\}. \tag{A.2}$$

In [46], the Gamma function was generalised to the multiple Gamma function $\Gamma_n$ satisfying the relations

$$\Gamma_{n+1}(z+1) = \frac{\Gamma_{n+1}(z)}{\Gamma_n(z)}. \tag{A.3}$$

Of these, the one of a particular interest for this paper is the double $\Gamma$ function $\Gamma_2$ which is closely related to the so-called Barnes $G$ function $G(z) = \Gamma_2^{-1}(z)$. It satisfies the recurrence identity

$$G(z+1) = \Gamma(z)G(z), \tag{A.4}$$

the initial condition

$$G(1) = 1, \tag{A.5}$$

and it can be shown to have the Weierstrass infinite product form [46]

$$G(z+1) = (2\pi)^{\frac{z}{2}} e^{-\frac{z(z-1)}{2} - \frac{\gamma z^2}{2}} \prod_{n=1}^\infty \left\{ \frac{\Gamma(n)}{\Gamma(z+n)} e^{z\psi(n) + \frac{z^2}{2}\psi'(n)} \right\}, \tag{A.6}$$

where $\psi$ denotes the logarithmic derivative of the Gamma function. The logarithm of the Barnes G function admits the integral representation [47,48]

$$\log G(z+1) = -\int_0^\infty \frac{dt\, e^{-t}}{t} \frac{\left\{ e^{-zt} + zt + \frac{z^2 t^2}{2} - 1 \right\}}{(1 - e^{-t})^2} - (1+\gamma)\frac{z^2}{2} + \frac{3}{2}\log\pi. \tag{A.7}$$

In this paper, we encountered an infinite product of $\Gamma$ functions in (3.63). Since this infinite product is convergent, it is reasonable to expect that it corresponds to a rational expression in finite products of Barnes $G$ functions. Here we consider the most general form of such an expression and lay out the necessary and sufficient conditions under which it leads to a convergent infinite product containing the Gamma functions alone. Then we compare it to the infinite products obtained in this paper (3.63, 3.65).

Let us consider the following expression in $G$ functions and its infinite product form due to (A.6).

$$\frac{G(1+u_1)\cdots G(1+u_p)}{G(1+v_1)\cdots G(1+v_q)}$$

$$= (2\pi)^{\frac{\mathbf{u}-\mathbf{v}}{2}} e^{-\frac{\mathbf{u}-\mathbf{v}}{2} - \frac{\gamma+1}{2}(\mathbf{u}^2 - \mathbf{v}^2)} \prod_{n=1}^\infty \Gamma^{p-q}(n) \frac{\Gamma(n+v_1)}{\Gamma(n+u_1)} \cdots \frac{\Gamma(n+v_q)}{\Gamma(n+u_p)} e^{(\mathbf{u}-\mathbf{v})\psi(n) + \frac{(\mathbf{u}^2 - \mathbf{v}^2)}{2}\psi'(n)}, \tag{A.8}$$

where $p$, $q$ are positive integers, and $\{u_1, \ldots, u_p\}$ and $\{v_1, \ldots, v_q\}$ are set of complex numbers not containing negative intgers $u_j, v_k \notin -\mathbb{N}$ ($j = 1, \ldots, p$, $k = 1, \ldots, q$). Here we have used the notations $\mathbf{u} = \sum_1^p u_j$, $\mathbf{v} = \sum_1^q v_j$, $\mathbf{u}^2 = \sum_1^p u_j^2$, and $\mathbf{v}^2 = \sum_1^q v_j^2$.

The infinite product on the right hand side is always convergent as it was obtained from a finite product of convergent products. For it to be an infinite product involving $\Gamma$ functions alone, the exponential terms in the product must have $\mathbf{u} = \mathbf{v}$ and $\mathbf{u}^2 = \mathbf{v}^2$.

In the following, we give two particular examples of this kind which are related to the results obtained in this paper:

1. $p = 4$, $q = 5$; $\Re(u_1) = \Re(u_2) = 0$, $u_1 = u_2^*$, $u_3 = -\frac{1}{2}$, $u_4 = 1$; and $v_1 = -\frac{1}{2} + u_1$, $v_2 = -\frac{1}{2} + u_2$, $v_3 = v_4 = v_5 = \frac{1}{2}$. The square of this is related to the result obtained in (3.64).

2. $p = 4$, $q = 8$; $\Re(u_1) = \Re(u_2) = -1$, $u_1 = u_2^*$, $\Re(u_3) = \Re(u_4) = 0$, $u_3 = u_4^*$; and $v_j = \frac{1}{2} + u_j$, $v_{p+j} = -\frac{1}{2}$ ($\forall$ $j = 1, \ldots, 4$). This is related to the the final result we obtained in (3.65). Note that these same conditions ensures that corresponding integral representation (3.66) is free of terms of the kind $z$ and $z^2$.

## B Thermodynamic limit of auxiliary functions

In this appendix we derive the thermodynamic limit for the ratios of Baxter polynomials (2.25) and transfer matrix eigenvalues (2.26).

We take the ratio of the Baxter polynomials $q_e(\lambda)$ to $q_g(\lambda)$ defined as (2.9) for an excited state and the ground state respectively

$$\phi(\lambda) \equiv \frac{q_e(\lambda)}{q_g(\lambda)} = \frac{\prod\limits_{j=1}^{N}(\lambda - \mu_j)}{\prod\limits_{j=1}^{\frac{M}{2}}(\lambda - \lambda_j)}. \tag{B.1}$$

For the excited state given by a two-spinon triplet $N = \frac{M}{2} - 1$, the logarithmic derivative of this function satisfies

$$[\log \phi]'(\lambda) = \int_{\mathbb{R}} d\tau \, \frac{\sigma_h(\lambda - \mu_{h_1}) + \sigma_h(\lambda - \mu_{h_2})}{\lambda - \tau}, \tag{B.2}$$

where $\sigma$ is the density satisfying integral equation

$$\sigma_h(\lambda) + \frac{1}{2\pi i} \int_{\mathbb{R}} d\tau \, K(\lambda - \tau) \sigma_h(\tau) = -\delta(\lambda). \tag{B.3}$$

This leads to the following asymptotic form in the thermodynamic limit for the ratio of Baxter polynomials $\phi(\lambda)$ for $\Im(\lambda) > 0$, $\Im(\lambda) < 0$ respectively

$$\phi(\lambda) = C_{\pm} \prod_{a=1}^{2} \frac{\Gamma\left(\pm \frac{\lambda - \mu_{h_a}}{2i}\right)}{\Gamma\left(\frac{1}{2} \pm \frac{\lambda - \mu_{h_a}}{2i}\right)}. \tag{B.4}$$

The constant can be fixed from the asymptotic behaviour of $\phi(\lambda) \sim \lambda^{-1}$ as $\lambda \to \infty$. This gives us

$$\phi(\lambda) = \begin{cases} \dfrac{1}{2i} \prod\limits_{a=1}^{2} \dfrac{\Gamma\left(\frac{\lambda - \mu_{h_a}}{2i}\right)}{\Gamma\left(\frac{1}{2} + \frac{\lambda - \mu_{h_a}}{2i}\right)}; & \Im(\lambda) > 0 \\[4ex] -\dfrac{1}{2i} \prod\limits_{a=1}^{2} \dfrac{\Gamma\left(-\frac{\lambda - \mu_{h_a}}{2i}\right)}{\Gamma\left(\frac{1}{2} - \frac{\lambda - \mu_{h_a}}{2i}\right)}; & \Im(\lambda) < 0 \end{cases}. \tag{B.5}$$

Similarly, we need to compute the ratio of the eigenvalues of the transfer matrix $\mathcal{T}$. This can be expanded in terms of the ratio of Baxter polynomials as

$$\chi(\lambda) \equiv \frac{\tau_e(\lambda)}{\tau_g(\lambda)} = \frac{1 + \mathfrak{a}_e(\lambda)}{1 + \mathfrak{a}_g(\lambda)} \frac{\phi(\lambda - i)}{\phi(\lambda)}. \tag{B.6}$$

When $\lambda$ becomes one of the root of the ground state $\lambda_a$, this becomes

$$\chi(\lambda_a) = M \frac{1 + \mathfrak{a}_e(\lambda_a)}{\mathfrak{a}'_g(\lambda_a)} \frac{\phi(\lambda_a - i)}{\phi'(\lambda_a)}, \tag{B.7}$$

where

$$\phi'(\lambda_a) = M \frac{\prod\limits_{j=1}^{\frac{M}{2}-1} (\lambda_a - \mu_j)}{\prod\limits_{\substack{j=1 \\ j \neq a}}^{\frac{M}{2}} (\lambda_a - \lambda_j)}. \tag{B.8}$$

The derivative $\mathfrak{a}'_g(\lambda_a)$ admits the thermodynamic limit

$$\mathfrak{a}'_g(\lambda_a) = -2\pi i M \rho_g(\lambda_a) + O(M^{-1}). \tag{B.9}$$

Also we have

$$1 + \mathfrak{a}_e(\lambda_a) = \left\{ \frac{1}{\phi(\lambda_a + i)} - \frac{1}{\phi(\lambda_a - i)} \right\} \phi(\lambda_a + i). \tag{B.10}$$

From (2.25), we can write the following identity in the thermodynamic limit

$$\frac{1}{\phi(\lambda \pm i)} = (\lambda - \mu_{h_1})(\lambda - \mu_{h_2}) \lim_{\epsilon \to 0^+} \phi(\lambda \pm i\epsilon). \tag{B.11}$$

On the other hand, we can show using the method introduced in [19] that although the ratio of Baxter polynomial on the real line is ill defined due to densification of its poles, the function $\phi'$ seen as the density of residues can be defined and it satisfies the identity

$$\frac{1}{M} \sum_{a=1}^{\infty} t(\pm(\lambda_a - \mu_k)) \phi'(\lambda_a) = \pm \phi(\mu_k - i). \tag{B.12}$$

Since $K(\lambda) = t(\lambda) + t(-\lambda)$, it leads to,

$$\frac{1}{M} \sum_{a=1}^{\infty} K(\lambda_a - \mu_k) \phi'(\lambda_a) = \phi(\mu_k - i) - \phi(\mu_k + i), \tag{B.13}$$

which gives us the following in the thermodynamic limit

$$\int_{\mathbb{R}} d\nu \, \rho_g(\nu) K(\lambda_a - \nu) \phi'(\nu) = \phi(\lambda_a - i) - \phi(\lambda_a + i). \tag{B.14}$$

Let us decompose the solution to this integral equation into parts which are analytic in one half of the complex plane and meromorphic on other as $\phi' = \phi'^{(+)} - \phi'^{(-)}$. We denote the part which is analytic in $\sigma \Im(\lambda) > 0$ by $\phi'^{(\sigma)}(\lambda)$. As it can be seen from (2.25), the function $\phi(\lambda - i\sigma)$ is analytic on $\sigma \Im(\lambda) > 1$, hence we automatically have a similar decomposition $\phi(\lambda - i) - \phi(\lambda + i)$ on the right hand side as well. Hence we can write

$$\int_{\mathbb{R}} d\nu \, K(\lambda - \nu) \rho_g(\nu) \phi'^{(\sigma)}(\nu) = \sigma \phi(\lambda - i\sigma), \quad \sigma = \pm. \tag{B.15}$$

It can be easily shown that for any function $f^{(\sigma)}$ which is analytic in $\sigma \Im(\lambda) > 0$ and integrable on real line $f^{(\sigma)}_{\mathbb{R}} \in L^1(\mathbb{R})$, the convolution with kernel $K$ produces shift

$$K * f^{(\sigma)}(\lambda) = 2\pi i f^{(\sigma)}(\lambda + i\sigma). \tag{B.16}$$

Hence by identifying $f^{\sigma}(\lambda) = \rho_g(\lambda)\phi'^{(\sigma)}(\lambda)$ we obtain

$$2\pi i\,\rho_g(\lambda)\,\phi'(\lambda) = \lim_{\epsilon\to 0^+}\{\phi(\lambda - i\epsilon) - \phi(\lambda + i\epsilon)\}. \tag{B.17}$$

Therefore, from (B.9–B.11) and (B.17), the ratio of eigenvalues of the counting function (B.7) can be written as

$$\chi(\lambda_a) = (\lambda_a - \mu_{h_1})(\lambda_a - \mu_{h_2})\phi(\lambda_a - i)\phi(\lambda_a + i). \tag{B.18}$$

Since $\chi$ is analytic function, we can extend it to all $\lambda \in \mathbb{R}$. Finally, from the thermodynamic limit of the ratio of Baxter polynomials $\phi$ obtained in (2.25), we get

$$\chi(\lambda) = \prod_{a=1}^{2}\tanh\left(\frac{\pi(\lambda - \mu_{h_a})}{2}\right). \tag{B.19}$$

Note that this result is also valid at the Bethe roots of the excited state $\chi(\mu_a)$. This can be either seen as a consequence of the analyticity or verified through the independent computations for $\chi(\mu_a)$ which follows a similar procedure as above.

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
