# Peer review of "Thermodynamic limit of the two-spinon form factors for the zero field XXX chain"

_SciPost Physics, doi:SciPost Phys. 6, 076 (2019)_

## Round 1 · Referee Report · Anonymous (Referee 1) · 2019-4-15

Strengths

  • exact results provided for a strongly interacting system

Weaknesses

  • lots of calculations with few discussions

Report

Very interesting paper that is able to show that the spinons form factors, derived in the context of q-vertex operator approach, can be also found by taking the thermodynamic limit of the lattice spin chain, in this case the XXX spin chain. Results are new and remarkable. I have some comment:

1) The derivation of the spinon form factor from the lattice theory was subject of intense research. Here the paper finally manage to accomplish this task thanks to a different representation of the Slavnov scalar product, the Foda-Wheeler expression, which allows for an easier thermodynamic limit. I believe this should be stressed both in the abstract and in the intro.

2) Form factors and their relation to correlation functions are not well presented. In the introduction there should be a discussion on what are the correlation function of interest and what is their representation in terms of spinons (their resolution of identity). On this regard also citations to these papers should be included Dugave et al, J. Stat. Mech. (2015) P05037 Dugave et al, J. Phys. A 49 (2016) 394001 since they have worked on the problem of correlation functions and spinons in the XXZ chain. Moreover the fact that finite temperature form factor representation of correlation functions have provided new results to finite temperature correlations in interacting integrable models is also presented in De Nardis, Panfil, 10.1088/1742-5468/aab012 De Nardis, Doyon, Bernard, arXiv:1812.00767 Cubero, Panfil, JHEP 104 (2019)

3) The large M behaviour of form factors of the ground states of 1d critical system is usually not trivial (M^\alpha with alpha non-integer decay). Here the author find M^-n decay, see 3.49. Is this expected? A clarification on this, especially clarifying relations with Luttinger liquid physics, is welcome.

4) Below 3.11 the authors claim that corrections 1/M to the matrix elements are not relevant. However the matrix is of size M so how to exclude that the corrections are relevant to the determinant?

5) Below 3.52 many claculations are presented without really explaining the logic and why they are done this way. More extensive explanations on this would be necessary.

6) A plot/figure of the Bethe rapidities on the complex plane would be also nice to display how spinons are written in terms of rapidities.

7) In eq. 2.33 the norm of the state seem to have been factorised in ground state contribution and spinon contribution, however this is not explained.

8) Please discuss how to extend the results to the XXZ chain in the conclusion.

Requested changes

see report

  • validity: high
  • significance: high
  • originality: high
  • clarity: good
  • formatting: good
  • grammar: good

Author:  Nikolai Kitanine  on 2019-04-15  [id 497]

(in reply to Report 1 on 2019-04-15)
Category:
answer to question

We would like to thank the referee for several useful comments, we would like also to reply to some of referee's questions and suggestions.

  1. The method described here can be applied for the XXX and XXZ case. The additional su(2) symmetry is a particularity of the XXX case and it brings some technical difficulties absent in the XXZ case. To deal with these difficulties we apply the Foda-Wheeler modification of the Slavnov formula, it permits us in particular to compute some integrals. We agree that this fact can be mentioned in the introduction, however we think that this purely technical part it is not that essential for this approach to be mentioned in the abstract.

  2. We thank the referee for raising this important question on the scaling behaviour of the form factors for zero and non-zero magnetic field. The non-integer scaling was obtained for the case of finite magnetic field and play an important role when the excitations (particles and holes) are close to the Fermi boundaries. In this paper we consider the zero magnetic field case (Fermi boundaries are at infinity in the thermodynamic limit) with holes in the bulk (far from the edges of the Bethe roots distribution for finite M). In this case it is easy to see that the boundary values of the shift function which give the non-integer scaling (see for example eq. (3.14) in N Kitanine et al J. Stat. Mech. (2011) P05028) are vanishing and lead to integer powers of M.

  3. Correction of order 1/M are essential for the computation of determinants of matrices of size proportional to M. For this reason these terms are carefully taken into account (holes contributions). We claim after eq. (3.11) that the corrections sub-leading with respect to 1/M can be neglected, it follows from the development of $det(I+xA)$ as power series in $x$.

  4. The denominator in eq. (2.33) contains two norms of two states involved in the form factor, there is no particular factorisation.

---

## Round 1 · Referee Report · Anonymous (Referee 2) · 2019-4-28

Strengths

1-The presented approach is tested against previously known results.
2-In principle, the approach can be generalized to other excited states.

Weaknesses

1-The approach uses the "bulk assumption" (see Report). It is a reasonable hypothesis but it has not been proved.
2-The computations will become more complicated for more complicated excited states.

Report

The authors present a new approach of computing form factors of integrable spin chains in the thermodynamic limit, based on the Algebraic Bethe Ansatz framework. The authors demonstrate the method on two-spinon form factors of spin operators in the XXX Heisenberg spin chain. Their final formula is in agreement with the previously known results of the q-vertex operator approach.

The approach of the authors can be summarized as the following: they write the determinant representation of the form factor (2.27) in the form (2.38) using a Foda-Wheeler version of the Slavnov determinant formula. The representation (2.38) is then evaluated as the product of the determinants of the "F-matrices" (3.1-3.2) corresponding to the ground state and the two-spinon excited state. These F-matrices are the product of an inverse Gaudin matrix and a Slavnov matrix. Their matrix elements are evaluated by writing the integral equations (3.8,3.20) and then evaluating these equations using the bulk assumption. The bulk assumption states that the tails of the distribution of real Bethe roots do not contribute to the leading order in the thermodynamic limit. It has not been proved yet.

Using the above analysis the authors obtain the intermediate result (3.26-3.29) for the form factor. The authors then proceed compute the determinants of the matrices (3.27-3.29). One of them is a Cauchy determinant, and the other is the determinant of a Cauchy matrix plus a finite rank matrix. Finally, the authors take the thermodynamic limit of the determinants using an infinite product factorisation. The final result of the form factor (3.64) is in agreement with previously known results in the q-vertex operator approach.

The presented approach could work for other low-lying excited states and other integrable models.

The paper is written in a clear and concise manner, presenting a new and scientifically relevant approach. I recommend it for publication.

Requested changes

I have found the following typos and minor mistakes, listed in order of appearance:
1-In the first sentence it should be "quantum integrable systems"
2-In the second paragraph of Sec. 2.3 it should be "excited state"
3-Before Eq. (2.35), there is a superfluous "a"
4-I suspect that the last term in the r.h.s. of formula (3.8) is present only if $\mu_k$ is real. However, the sentence after (3.8) suggests that $\mu_k$ is not always real.
5-After Eq. (3.9), $\rho(\lambda,\mu)$ should be $\rho_g(\lambda,\mu)$
6-In the r.h.s. of Eq. (3.22), $\mu$ should be $\mu_{h_a}$
7-After formula (3.58), it should be "infinite product"

---

## Round 1 · Referee Report · Frank Göhmann (Referee 3) · 2019-5-5

Strengths

1.) provides a link between two complementary methods
2.) technical part is written in a well-understandable way

Weaknesses

1.) Introduction could place the work better into context of current research

Report

Based on a certain assumption, called "the bulk assumption" in the text, the authors re-derive, within the framework of the algebraic Bethe Ansatz, the well-known and useful formula for the square of the two-spinon form factor of a local operator in the Heisenberg spin chain. This formula had previously been obtained within the so-called q-vertex operator approach [2] which, loosely speaking, is a variant of the algebraic Bethe Ansatz working directly on the infinite lattice. Hence, the q-vertex operator approach has the advantage that the thermodynamic limit is already build in. Its main disadvantage is that this method does not allow for the inclusion of external
fields and, at first instance, is restricted to models with finite
mass gap. Results for massless models, such as the isotropic Heisenberg chain, were obtained by taking massless limits. The algebraic Bethe Ansatz method, on the other hand, is applicable for massive and massless phases and also at finite external fields [3,14]. Here the problem is to properly take the thermodynamic limit.

So far the latter problem was considered for the anisotropic
Heisenberg (or XXZ) chain at finite magnetic field in

1.) N. Kitanine, K.K. Kozlowski, J.-M. Maillet, N.A. Slavnov, and V. Terras, "On the thermodynamic limit of form factors in the massless XXZ Heisenberg chain", J. Math. Phys. 50 (2009), 095209

2.) K.K. Kozlowski, "Form factors of bound states in the XXZ chain", J. Phys. A 50 (2017), 184002

or in the massive regime

3.) M. Dugave, F. Göhmann, K.K. Kozlowski, and J. Suzuki,
"On form factor expansions for the XXZ chain in the massive
regime", J. Stat. Mech. 1505 (2015), P05037.

The expressions for the squares of the form factor densities derived in the above references differed from those obtained within the vertex operator approach in that the former contain certain Fredholm determinants. Numerical agreement of the two types of expressions for the XXZ chain in the massive regime was demonstrated in 3.).

In the present paper the authors perform the thermodynamic limit directly for the Heisenberg chain at zero magnetic field and obtain the densities of the squared form factors in the form known from the vertex operator approach. This provides once more strong evidence that the two methods lead to the same results.

The paper is interesting for experts working in the field not because of the final formula, that was known in an equivalent form, and not even because of the fact that it establishes equivalence of the two approaches, but rather because of certain methodological aspects in its bulk. First there is the use of the Foda-Wheeler variant of Slavnov's formula which here proves to be useful for the first time. Second, the Cauchy determinant extracted in section 3.2 corresponds to the solution of the Lieb equation and is not the Cauchy determinant extracted in previous cases. Third, the way the thermodynamic limit is finally taken in equation (3.52), using the second functional equation
for the gamma function along with its Weierstrass product representation is something interesting I have not seen before.

The paper seems technically correct. The material in the bulk is
well-presented and easy to follow, at least for experts. The only
delicate point is the "bulk assumption" mentioned in the beginning. Although the calculation based on this assumption leeds to a known result, it seems rather non-obvious to me and certainly deserves further attention in future studies. It should be possible to justify (or falsify) this assumption by studying the non-linear integral equation for the auxiliary functions $\mathfrak{a}_g$ and and $\mathfrak{a}_e$.

Altogether I would like to recommend the paper for publication.

Here is a list of points that, to my opinion, might help to make
the paper more accessible, especially to non-expert readers.

p 5: "It is well-known that the ground state [...] of the XXX
chain is the only ...": If well-known then provide a reference.

p 5: "... the only zeros of the function $\mathfrak{a}_g (\lambda) + 1$ are the Bethe roots and they are all real ...": As can be seen from the second equation (2.10) this functions has more zeros, the zeros of $\tau(\mu)$. What you probably mean is: "The only real zeros of ...".

p 9: "... with the similar arguments, one can also show that ...": Maybe better say something using the words "selection rule due to the conservation of the total spin".

p 10: The reasoning leading from equation (3.1) to (3.10) is
almost literally taken from Appendix A of the paper "Integral
representations for correlation functions of the XXZ chain at finite temperature", J. Phys. A37 (2004) 7625-7652 by F. Göhmann, A. Klümper and A. Seel, which may deserve to be cited here, perhaps in conjuction with [22].

p 10: As a remark: The point $\mu = {\rm i}/2$ does not need a separate calculation if the contour Gamma is defined accordingly. Then $\mu = {\rm i}/2$ is allowed in (3.13).

p 12: $\mu \rightarrow \mu_{h_a}$

p 15: incircles $\rightarrow$ encircles

p 20: "This formula is the finite result ..." $\rightarrow$ "This formula is the final result ..."

Reference section: [7] heisenberg $\rightarrow$ Heisenberg,
[31] bethe $\rightarrow$ Bethe, [33]: Title should be
``Zur Theorie der Metalle I. Eigenwerte und Eigenfunktionen
der linearen Atomkette'', [17]: This is on XXZ. I guess the
authors would rather have wanted to cite the paper M. Karbach
et al., Phys. Rev. B 55 (1997), p 12510.

Maybe finally some comments on the introduction are in order. In general I think the authors missed out a chance to put their work into a larger context. There has been quite some activity in the last ten years or so on the calculation of form factors in
lattice models and in integrable quantum field theories which
might have been mentioned and put into relation with the submitted work. Such an effort would have been beneficial for the less specialized reader.

A reference which I would cite in the introduction together with [23] is the paper of M. Lashkevich, "Free field construction for the eight-vertex model: representation for form factors", Nucl. Phys. B 621 (2002) 587, as it lays the technical foundations for [23].

Requested changes

see attached report

---

## Round 2 · Referee Report · Frank Göhmann (Referee 3) · 2019-5-21

Report

I think there is not much to add to my previous report. The authors have further improved the manuscript by taking account of the referee's suggestions. The manuscript should now be published as it stands.

---

## Round 2 · Referee Report · Anonymous (Referee 2) · 2019-6-4

Report

In my previous report, I have already recommended the paper for publication. With the improvements of this version, I recommend the paper for publication as it is.

---

## Round 2 · Referee Report · Anonymous (Referee 1) · 2019-6-7

Report

I remain of the idea that the paper is still too technical and that a more general introduction on spinons and form factors expansion of correlation functions was needed. But overall I think that now the paper is ready for publication.

---

## Round 2 · Author Response

We would like to thank the referees for several useful comments and also for the detected typos, we followed most of the referees suggestions in the new version. We would like also to reply to some of questions and suggestions.

Referee 1:

  1. The method described here can be applied for the XXX and XXZ case. The additional $\mathfrak{su}(2)$ symmetry is a particularity of the XXX case and it brings some technical difficulties absent in the XXZ case. To deal with these difficulties we apply the Foda-Wheeler modification of the Slavnov formula, it permits us to compute some integrals and helps a lot for the computation. We agree that this fact can be mentioned in the introduction, however we think that this purely technical part it is not that essential for this approach to be mentioned in the abstract.

  2. We thank the referee for reminding us some extra papers on thermal form factors, in particular Dugave et al, J. Stat. Mech. (2015) P05037 is highly pertinent to the present paper (due to the discussion on the comparison of $q$-vertex operator and ABA results). We are also aware of the large literature on the form factors for integrable quantum field theories, but we think that this paper is dealing only with the spin chains and we have mostly restricted our reference list to these systems.

  3. We thank the referee for raising this important question on the scaling behaviour of the form factors for zero and non-zero magnetic field. The non-integer scaling was obtained for the finite magnetic field case and it plays an important role when the excitations (particles and holes) are close to the Fermi boundaries. In this paper we consider the zero magnetic field case (Fermi boundaries are at infinity in the thermodynamic limit) with holes in the bulk (far from the edges of the Bethe roots distribution for finite $M$). In this case it is easy to see that the boundary values of the shift function which give the non-integer scaling (see for example eq. (3.14) in N. Kitanine et al J. Stat. Mech. (2011) P05028) are vanishing and lead to integer powers of $M$. We agree that this discussion can be included in the paper.

  4. Corrections of order $1/M$ are essential for the computation of determinants of matrices of size proportional to $M$. For this reason these terms are carefully taken into account (holes contributions). We claim after eq. (3.11) that all the corrections sub-leading to $1/M$ can be neglected, it follows from the development of $\det(I+xA)$ as a power series in $x$.

  5. We added some comments explaining the procedure.

  6. We are not sure that a picture (in the simplest two-spinon situation) would help much in this context.

  7. The denominator in eq. (2.33) contains two norms of two states involved in the form factor, there is no particular factorisation.

Referee 2:

We agree that the bulk assumption is the most subtle point of the presented method and for this reason we wanted first test it by obtaining a well established result. We also agree with Frank Göhmann that the proof of this assumption (or a weaker one explaining why some finite size corrections are not pertinent in the thermodynamic limit) could be tried through the analysis of non-linear integral equations. However this analysis goes far beyond the scope of the present paper.

Referee 3:

We would like to thank Frank Göhmann for several important remarks, in particular for reminding us to cite the paper [45] (in the new version) where effectively a very similar approach is used to compute the ratio of two matrices.

We have deliberately chosen a rather short introduction for this (rather technical) paper covering the simplest form factor reserving a more detailed analysis of the context for forthcoming publications treating states involving complex roots. However we have added some important references which were evidently missing.

---

## Round 2 · List of Changes

1. Introduction: slightly extended, several references added as suggested by referees
2. Section 3: citation of the paper [45] is added (as suggested by referee 3), some explanation on the scaling behaviour is provided at the end of subsection 3.2 (as suggested by referee 1), some explanation of the computations are given after eq. (3.52)
3. One acknowledgment added.

Several typos corrected throughout the paper

---

## Editorial Decision

published